# Two-dimensional ferroelectric channel transistors integrating ultra-fast memory and neural computing

Shuiyuan Wang[1], Lan Liu[1], Lurong Gan[1], Huawei Chen[1], Xiang Hou[1], Yi Ding[1], Shunli Ma[1], David Wei Zhang[1] & Peng Zhou [1✉]

With the advent of the big data era, applications are more data-centric and energy efficiency issues caused by frequent data interactions, due to the physical separation of memory and computing, will become increasingly severe. Emerging technologies have been proposed to perform analog computing with memory to address the dilemma. Ferroelectric memory has become a promising technology due to field-driven fast switching and non-destructive readout, but endurance and miniaturization are limited. Here, we demonstrate the α-In$_2$Se$_3$ ferroelectric semiconductor channel device that integrates non-volatile memory and neural computation functions. Remarkable performance includes ultra-fast write speed of 40 ns, improved endurance through the internal electric field, flexible adjustment of neural plasticity, ultra-low energy consumption of 234/40 fJ per event for excitation/inhibition, and thermally modulated 94.74% high-precision iris recognition classification simulation. This prototypical demonstration lays the foundation for an integrated memory computing system with high density and energy efficiency.

---

[1] ASIC & System State Key Lab., School of Microelectronics, Fudan University, Shanghai 200433, China. ✉email: pengzhou@fudan.edu.cn

The rise of artificial intelligence has led to explosive growth in emerging data-centric applications represented by images recognition and classification[1,2]. Data-intensive tasks require computing systems to perform batch parallel processing, frequently accessing results and interacting from memory domains[3]. The computing and memory components of modern computers are physically separated[4], and massive communication increases unexpected power consumption and degrades efficiency, causing the so-called von Neumann bottleneck[5].

Emerging memory devices such as memristors[6], memtransistors[7], phase change memory[8], electrical double-layer transistors[9], two-dimensional (2D) heterojunction devices[10], and ferroelectric field effect transistors (FeFETs)[11] are used to perform analog computation in an attempt to break out of the dilemma. FeFETs with switchable electric dipoles, fast operation[12,13] and non-destructive readout[14] are ideal for building low-power[12], high-efficiency memory computing integrated systems. However, traditional FeFETs use ferroelectrics as the dielectric layer to modulate channel conductance[14]. The residual polarization decreases with cumulative switching cycles, and ferroelectric fatigue occurs[15], resulting in memory with limited endurance[16].

In addition, the future trend of high-integration artificial intelligence applications is pushing the design of memory and computing elements toward miniaturization[17,18]. Bulk perovskite[19], oxide[11,20], or organic ferroelectric polymer[21] are served as the gate dielectric in conventional FeFETs, which is insufficient for continuous scaling both in vertical and planar dimensions[22]. 2D layered semiconductors with atomic thickness possess the potential for continuous shrinking[23–25], which is a promising candidate for future high-density memory and computing systems[26,27]. Particularly, 2D layered α-In$_2$Se$_3$ exhibits robust ferroelectricity at room temperature (RT) without annealing[14,28,29], and thanks to the intrinsic interlocking of dipoles in α-In$_2$Se$_3$[18,22], it can maintain ferroelectric polarization even at atomic scale.

Here, distinct from the conventional FeFETs, 2D ferroelectric semiconductor α-In$_2$Se$_3$ was exploited as the channel materials to demonstrate a compact scalable device that integrates non-volatile memory (NVM) and neural computing functions. 2D α-In$_2$Se$_3$ ferroelectric channel transistors (FeCTs) show absorbing performance, including NVM large memory hysteresis windows, long-term retention, enhanced endurance by internal electric field, fast write speed of 40 ns, flexible adjustment of neuroplasticity and ultra-low power consumption of 234/40 fJ per event for excitation/inhibition. Moreover, 2D FeCTs exhibit thermal tunability in both memory and neural computation, and based on FeCTs, a simulated iris recognition and classification with the best accuracy of 94.74% comparable to the software is realized. The elaborate prototype devices pave the way for building high-density, energy-efficient memory and computing fusion systems, providing promising candidates for eliminating the physical separation of memory and computing.

## Results

Figure 1a shows the schematic of 2D FeCTs, which integrates memory and computing capabilities, that is, non-volatile memory and neural computing. 30 nm Al$_2$O$_3$ is deposited on the substrate by atomic layer deposition (ALD) as the bottom dielectric layer, and the bottom h-BN was prepared by mechanical exfoliation for interface optimization, followed by the transfer of the exfoliated 2D α-In$_2$Se$_3$ channel layer. Then we transfer h-BN as the top dielectric layer, electron beam lithography forms the electrode pattern, electron beam evaporation deposits the source drain and the top gate (TG), while heavily p-doped silicon substrate as the global gate (GG) (detailed process flow shown in Supplementary Fig. S1 in Supplementary Information). Figure 1b shows the false

color atomic force microscope (AFM) amplitude error image of the 2D FeCTs. And the detailed thickness information of α-In$_2$Se$_3$, bottom h-BN and top h-BN measured by AFM is provided in Supplementary Fig. S3a in Supplementary Information, which are approximately 40 nm, 20 nm and 20 nm, respectively. In addition, the high-resolution scanning transmission electron microscope (STEM) was used to characterize and observe the 2D FeCTs microstructure. The cross-sectional image and corresponding energy dispersive X-ray spectroscopy (EDS) element mapping at the interface between the multilayer h-BN and α-In$_2$Se$_3$ are shown in Fig. 1c, which implies a clean van der Waals heterojunction with negligible interface state trapping effect. Figure 1d shows the Raman spectrum of the channel α-In$_2$Se$_3$ to characterize the material properties, which is consistent with previous reports[14,18,30]. It is worth noting that the additional splitting peak near $90\ cm^{-1}$ can be regarded as an indication of hexagonal (2H) stacking[18]. To further confirm the 2H structure, we have supplemented the X-ray diffraction (XRD) characterization of α-In$_2$Se$_3$ crystals, as shown in Supplementary Fig. S2. The diffraction pattern only shows the $c$-plane peak and its higher-order interplanar spacing. The peak pattern can determine the lattice constant $c$ ($\approx$19.23 Å), which is consistent with the reported 2H α-In$_2$Se$_3$, and is significantly different from 3 R α-In$_2$Se$_3$[31–33]. And the Raman characterization of h-BN is shown in Supplementary Fig. S3b in Supplementary Information. Crucially, we transferred the 40 nm channel α-In$_2$Se$_3$ ferroelectrics onto a conductive Au/Al$_2$O$_3$ substrate and determined its ferroelectric polarization by piezoelectric microscopy (PFM). Figure 1e shows the three-cycle off-field PFM amplitude hysteresis loop of the 40 nm channel α-In$_2$Se$_3$, and the inset includes a schematic of PFM test structure and the PFM phase hysteresis loop (the on-field PFM amplitude and phase hysteresis loops are shown in Supplementary Fig. S4), which indicates a significant ferroelectric polarization flip. Figure 1f records the phases of the outer rectangular track and inner square scanned on the same α-In$_2$Se$_3$ by PFM domain engineering with applying +8 V and −8 V bias to the conductive probe respectively. The strong PFM phase contrast in the electrical writing region visually shows the polarization reversal occurring in the channel α-In$_2$Se$_3$.

First, we compare traditional FeFETs with 2D α-In$_2$Se$_3$ FeCTs, as shown in Fig. 2a, b. It is worth emphasizing that there is only polarization bound charges in the FeFETs[14], which is reversed to achieve polarization switching, and finally modulate channel conductance (Fig. 2a). In contrast, 2D α-In$_2$Se$_3$, as ferroelectric semiconductors[34], have mobile charges as the nature of semiconductor in addition to polarized bound charges[14]. When 2D α-In$_2$Se$_3$ FeCTs in the polarization-up state, positive and negative polarized bound charges are distributed on the upper and lower surfaces of the channel, respectively, shown in Fig. 2b. Therefore, positive and negative movable charges are accumulated on the top and bottom surfaces of the channel, forming an upward built-in electric field to maintain the polarization-bound charge, which indicates long retention time and improved endurance of NVM. Next, we discuss the working mechanism. For convenience, we omit the interface optimization h-BN, and only discuss the band bending caused by polarized bound charges in the channel, without considering the mobile charges, which is consistent with the complete theory considering both ferroelectric and semiconductor properties[14]. Note that the gate dielectric of fabricated 2D FeCTs has a high equivalent oxide thickness (EOT) and the electric field along the channel is not strong enough, resulting in incomplete polarization switching and localized mobile charges distribution. GG applies a negative voltage (below coercive voltage), the bottom channel distributes positive polarization bound charges, the energy band bends downward, and accumulates electrons, which forms a low resistance state (LRS "1"),

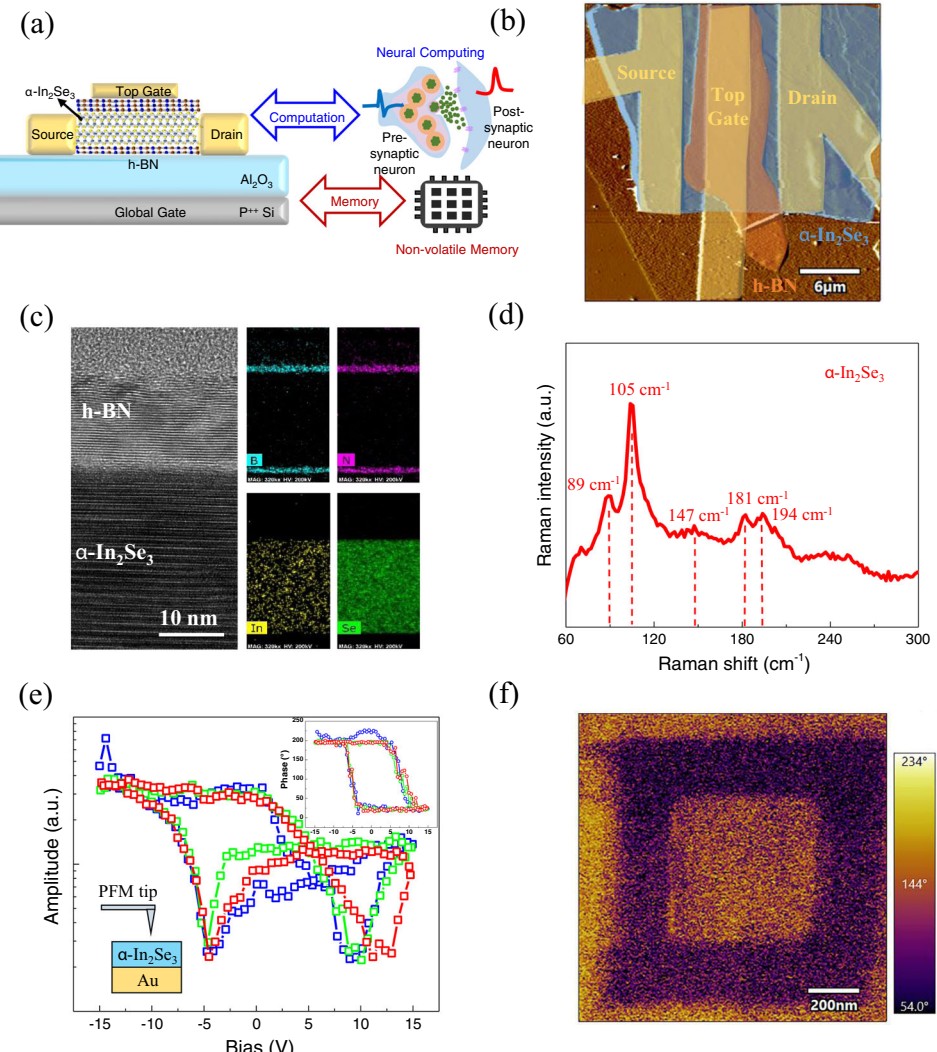

**Fig. 1 Schematic and characterization of 2D α-In₂Se₃ FeCTs for memory and computing. a** The 2D FeCTs structure integrates memory and computing functions, in which α-In₂Se₃ ferroelectrics serve as channel layer, Al₂O₃ and hBN as the global and top dielectric layers respectively. Non-volatile memory is implemented by GG, and neural computing is realized simultaneously with combination of TG. **b** False color AFM amplitude error image of the 2D FeCTs. Sacle bar: 6 μm. **c** High-resolution STEM image of multilayer h-BN and α-In₂Se₃ heterojunction and corresponding EDS element mapping in FeCTs, which indicates a clean interface. Sacle bar: 10 nm. **d** Raman characterization of α-In₂Se₃, where the 89 cm⁻¹ peak position suggests a hexagonal structure. **e** The three-cycle off-field PFM amplitude hysteresis loop of the channel α-In₂Se₃, the illustration includes the test structure and the PFM phase hysteresis loop, which shows a clear ferroelectric polarization flip. **f** The phase image after writing the outer rectangular track and inner square by applying +8 V and −8 V bias to the α-In₂Se₃ through PFM. Sacle bar: 200 nm.

corresponding to the polarization down (Fig. 2c). Conversely, GG applies a positive voltage (above coercive voltage), the negative polarization bound charge is distributed at the bottom of channel, energy band bends upward, and the electrons are depleted, which results in a high resistance state (HRS "0") corresponding to polarization up (Fig. 2d). The working mechanism of TG is similar to GG as exhibited in Fig. 2e, f, but it is worth mentioning that the coverage area of the gate to the channel directly affects the modulation performance. Although the top dielectric (h-BN) has a lower EOT and induces a stronger electric field, the top electric field does not completely cover the channel, resulting in a weaker polarization of the channel ferroelectric and depolarization in a short time. Channel conductance returns to its initial state, showing short-term plasticity, and negative voltage shows conductance increase (LRS "1"), corresponding to short-term potentiation. However, with the accumulation of negative voltage pulses, the channel ferroelectric polarization is strengthened, and finally a non-volatile

polarization is formed, that is, long-term potentiation, and vice versa. In this way, 2D FeCTs show the coexistence and evolution of volatile and nonvolatile, which is exactly what neural computing expects to simulate the short- and long-term plasticity in biology[3,6]. The energy band of 2D α-In₂Se₃ FeCTs under dynamic equilibrium is shown in Supplementary Fig. S5.

Figure 3a shows the 2D α-In₂Se₃ FeCTs transfer curves under varying bidirectional scanning voltages, where the source-drain voltage (Vds) is fixed at 1 V, and the clockwise hysteresis memory windows enlarge with the incremental GG voltage (VGG) sweeping, showing the accumulation of ferroelectric channel polarization switching (Supplementary Fig. S6a exhibits the output curves under varying VGG). Figure 3b extracts the memory windows, showing a maximum window of 6 V under −8 to 8 V sweeping. Under sufficient GG write and erase spikes (±8 V, 10 s and read at VGG = 0 V, Vds = 1 V), the retention characteristics of FeCTs show stable non-volatile, with high and low states exceeding 10³, as shown in Fig. 3c. The basic erase (LRS) and

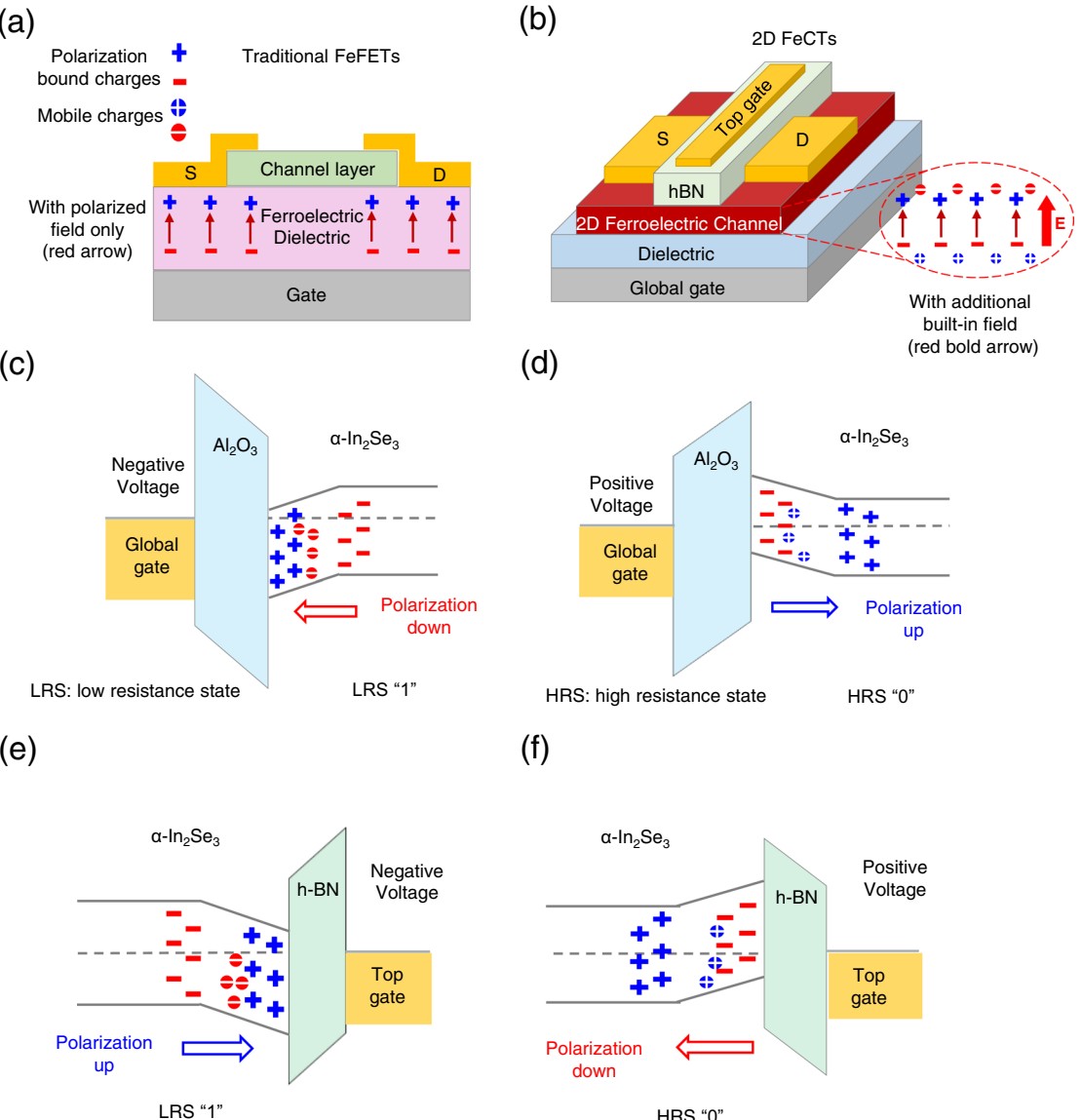

**Fig. 2 Comparison of FeFETs and FeCTs structure and working mechanism. a** For FeFETs, traditional ferroelectrics act as the dielectric layer, where only polarization-bound charges exist, and channel conductance is modulated by polarization switching. **b** The ferroelectric semiconductor 2D α-In$_2$Se$_3$ in FECTs serves as the channel layer and has both polarized bound charges and mobile charges. When in a polarized upward state, positive and negative polarized bound charges are distributed on the top and bottom surfaces of the channel, respectively, which causes positive and negative mobile charges to accumulate in opposite positions, forming an upward built-in electric field to maintain upward polarization. **c** For GG, a negative voltage lower than the coercive voltage is applied, and the bottom channel distributes positive polarized bound charges, the energy band is bent downward, and electrons are accumulated to form LRS, which corresponds to downward polarization. **d** When GG applies a positive voltage higher than the coercive voltage, the negative polarized bound charges are distributed at the bottom of the channel, and the energy band bends upward. The depletion of electrons results in HRS, which corresponds to the upward polarization. **e** Similarly, for TG, a negative voltage is applied, the top channel distributes positive polarized charges, and electrons accumulate to form LRS, which corresponds to polarization up. **f** A positive voltage applied in TG, the top channel distributes negative polarized bound charges, and electrons are depleted to form HRS, corresponding to polarization down. It should be noted that owing to the high EOT of FeCTs gate dielectric and the insufficient electric field along the channel, incomplete polarization switching and localized mobile charges distribution are caused.

write (HRS) characteristics of NVM are described in Fig. S7a. Thanks to the nature of α-In$_2$Se$_3$ semiconductors, the existence of mobile charges creates a built-in electric field, which consolidates and strengthens the polarization of the ferroelectric dipole and improves the endurance. Figure 3d depicts the endurance of α-In$_2$Se$_3$ FeCTs at 500 erase and write cycles, showing negligible HRS and LRS degradation (the dynamic erase/write response is shown in Fig. S7b). In addition, we explored the programming speed of FeCTs NVM in fixed LRS. The LRS-HRS ratio decreases

as the write width decreases, but it is worth noting that 40 ns can still be effectively written, as shown in Fig. 3e, which is consistent with the ultra-fast switching of ferroelectric polarization[12,13]. Further, Fig. 3f shows that after applying a +8 V, 40 ns ultrafast write spike, the channel transitions from the initial LRS "1" to HRS "0" (approximately an order of magnitude) and is non-volatile, which proves the ultra-fast programmability of 2D α-In$_2$Se$_3$ FeCTs NVM, and the insert is the actual waveform of the ultrafast write spike.

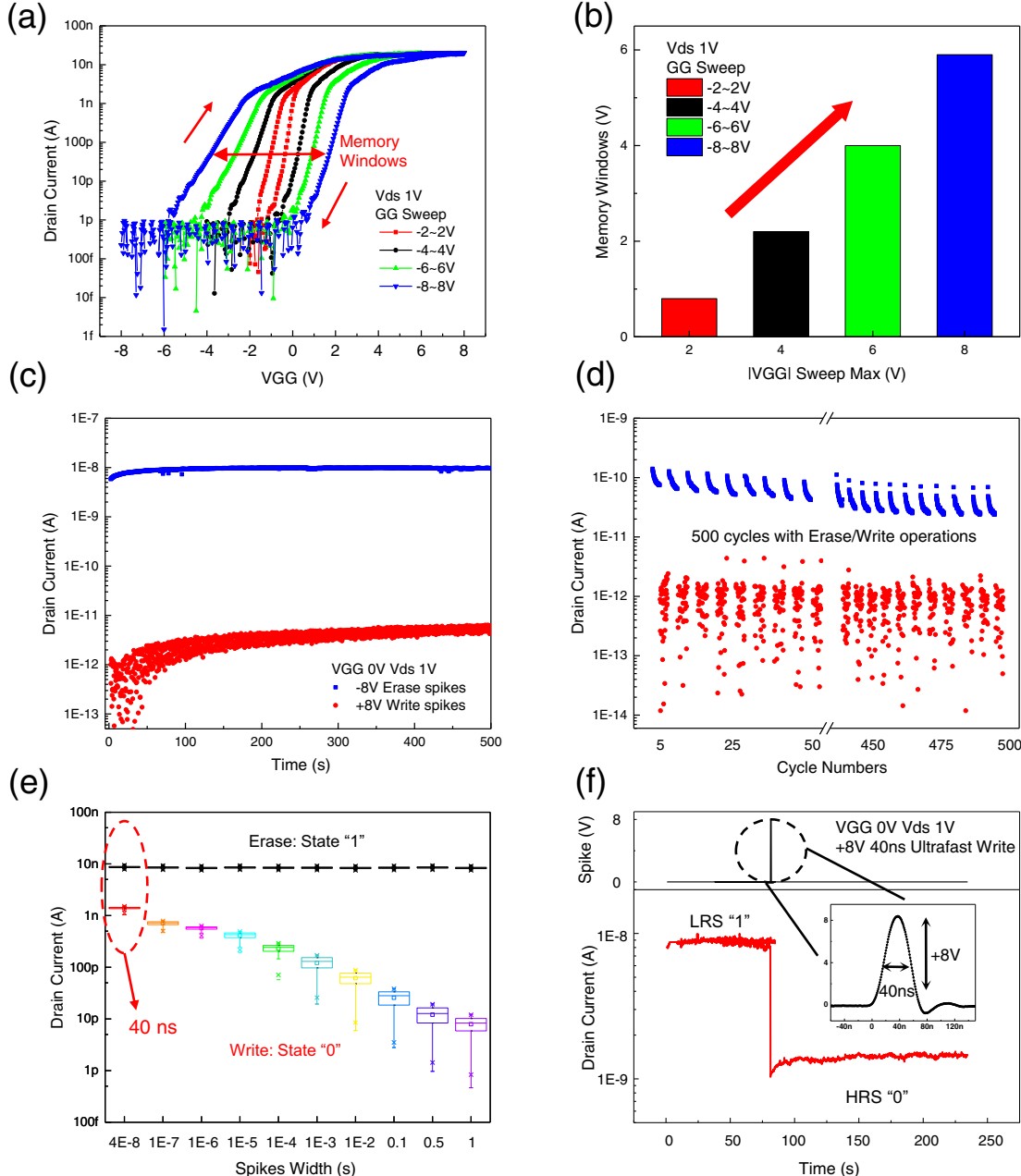

**Fig. 3 2D α-In₂Se₃ FeCTs for NVM with ultrafast writing. a** Transfer curves of 2D FeCTs NVM, where the clockwise hysteresis windows expand with increasing VGG, showing cumulative channel polarization. **b** The memory window extracted in (a) is incremented to get a maximum of 6 V. **c** NVM retention characteristics, with sufficient GG write and erase spikes (±8 V, 10 s and read at VGG = 0 V, Vds = 1 V), LRS/HRS ratio exceed 10³. **d** The robust endurance of α-In₂Se₃ FeCTs NVM after 500 erase and write cycles, showing negligible degradation of HRS and LRS. **e** FeCTs NVM programming speed with fixed LRS. LRS/HRS ratio decreases with the decrease of the writing width, and even 40 ns can implement effective writing, which may be attributed to the ultra-fast ferroelectric switching. **f** Under the action of a 40 ns (+8 V) write spike, the channel current transitions from the initial LRS "1" to HRS "0" and shows non-volatile data retention, which confirms the ultrafast programmability of 2D α-In₂Se₃ FeCTs NVM. The inset is an actual waveform of 40 ns ultrafast write spike.

Next, we explored 2D α-In₂Se₃ FeCTs for neural computing, where TG voltage (VTG) spikes simulate presynaptic input and ferroelectric channel current is monitored as post-synaptic current (PSC). TG applies short negative spikes (−8~−5 V, step 1 V), and with the increase of the spike amplitudes, the FeCTs exhibit incremental PSC variations in response to the spikes, but can return to the initial state quickly, which simulates a typical biological short-term plasticity (STP)[10,35], as shown in Fig. 4a. TG output and transfer curves are displayed in Supplementary Figs. S6b and S8a respectively in Supplementary Information.

Besides, Fig. S9 investigated the inhibitory PSC induced paired pulse facilitation (PPF) characteristics of STP, which gradually recover to 100% as the spike interval increases, and is described in detail in Supplementary Information. Figure 4b depicts the spike-rating-dependent plasticity (SRDP) of FeCTs neural computing, where the SRDP gain is proportional to the stimulation frequency. Specifically, for relatively high-frequency stimulation, it shows a strong inhibitory effect (11.11 Hz inserted in Fig. 4b), and for relatively low-frequency spike input, the inhibitory gain is weak (0.1 Hz inserted in Fig. 4b). Moreover, simulation of long-

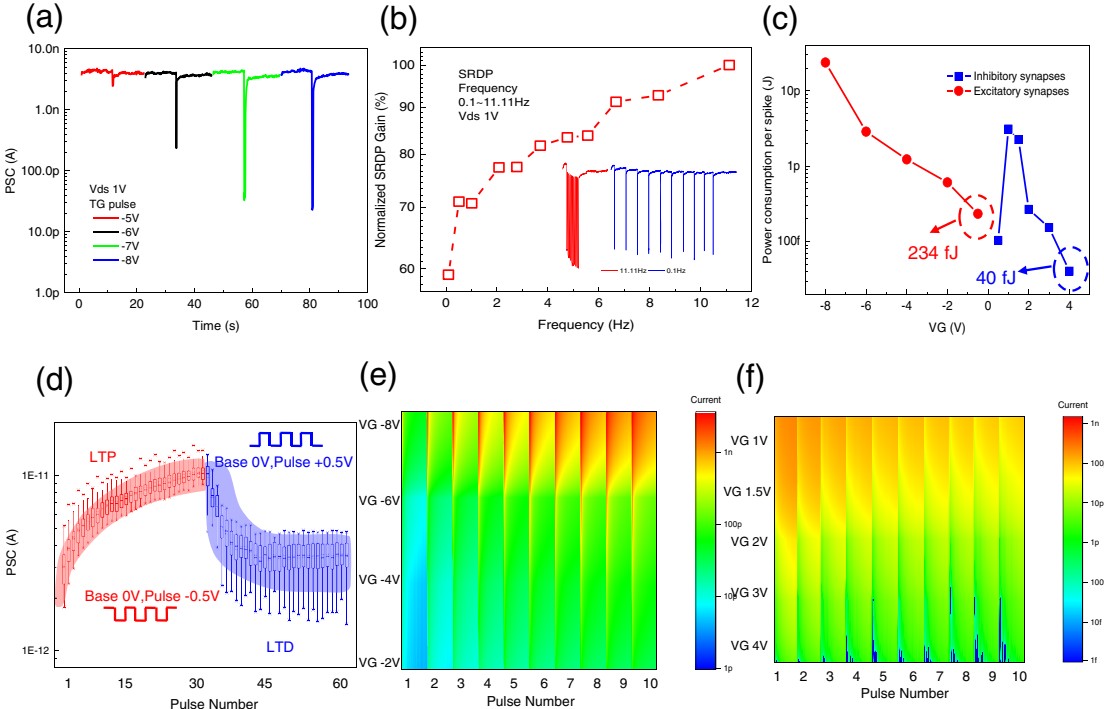

**Fig. 4 Neural computation of 2D α-In$_2$Se$_3$ FeCTs, including properties available for spiking and artificial neural networks. a** Under short negative spikes (−8 ~ −5 V with TG step of 1 V), FeCTs exhibit incremental PSC amplitude, which strengthens with the increase of the spike amplitude, and can be refreshed quickly, thus simulating a typical neural STP. **b** SRDP in FeCTs neural computation, where the SRDP gain is proportional to the stimulation frequency. For relatively high-frequency stimuli (11.11 Hz in the illustration), it shows a strong inhibition effect, while for relatively low-frequency spikes (0.1 Hz in the illustration), the gain is weak. **c** Calculated single event energy consumption of FeCTs for simulated excitatory and inhibitory synapses under varying VG stimuli. 2D FeCTs show ultra-low power consumption, including the minimum excitation/inhibition of 234/40 fJ per spike. **d** Progressive excitatory and inhibitory PSC modulation is realized with ultralow spike voltages ( ± 0.5 V, 30 ms), corresponding to the LTP and LTD simulation. **e** PSC mapping with 10 consecutive spikes with varying negative VG. Excitatory PSC was obtained under negative VG, and the current increased with amplitude and spike accumulation, showing LTP behavior. **f** PSC mapping of 10 consecutive varying positive VG spikes. Inhibiting PSC appears under positive spikes, and current decreases with increasing amplitude and spikes, showing LTD behavior.

term plasticity in neuromorphic engineering can be implemented in FeCTs as well. Minimal spikes of ± 0.5 V are applied to the gate to achieve a progressive excitatory and inhibitory PSC modulation (the original PSC curves see Supplementary Fig. S8b in Supplementary Information), corresponding to long-term potentiation (LTP) and long-term depression (LTD)[10], respectively, as statistical analyses in Fig. 4d (The box chart contains data ranging from 25 to 75%, with the cross at the top and bottom representing 99 and 1% of the data, respectively. The upper and lower horizontal lines represent the maximum and minimum values of the data, and the rectangle represents the mean value of the data). Excitation and inhibition modulation implemented with minimal spike voltage, which implies ultra-low energy consumption of 2D FeCTs for neural computing. Intuitively, we plot the PSC mapping of 10 consecutive spikes with varying gate voltage (VG). Figure 4e indicates that an excitatory PSC is obtained under the negative gate spikes, and the current increases with amplitude and spike accumulation, showing LTP behavior. Conversely, Fig. 4f implies an inhibitory PSC under positive spikes, and the current decreases with increasing amplitude and spikes, exhibiting LTD behavior (Note that the current in the PSC mapping does not include all voltage ranges, but to show the evolution trend more vividly and intuitively). Furthermore, we calculated the energy consumption per spike of FeCTs simulated excitatory and inhibitory synapses under different gate voltage stimuli, as shown in Fig. 4c. For excitatory synapses, single spike energy expenditure decreases monotonically. As for inhibitory synapses, the increase in voltage amplitude leads to a decrease in

the response current, and at higher voltage amplitudes, current degradation dominates, which makes the single-spike energy consumption increase first and then decrease. It is impressive that 2D α-In$_2$Se$_3$ FeCTs generally exhibit lower power consumption, especially for analog inhibitory synapses, even as low as 40 fJ per spike, which makes them promising candidates for energy-efficient neuromorphic systems. And it is worth noting that performing neural computation will have some impact on the memory performance, but will not cause NVM failure, and the memory functions can still be implemented reliably.

Surprisingly, 2D α-In$_2$Se$_3$ FeCTs show flexible thermal tunability both for memory switching and neuromorphic computing. We first measured basic electrical characteristics, where Supplementary Fig. S10a shows the output characteristics (Vds from −1 to 1 V, VGG and VTG are fixed at 0 V) as a function of thermal temperature (298~423 K). As the temperature increases, the current climbs and a better contact is formed, which may be attributed to thermally affected channel ferroelectric polarization[18] and defect healing[36]. The thermal temperature-dependent transfer curves (VGG = −8~8 V, Vds = 1 V) are shown in Supplementary Fig. S10b. Similarly, it still has clear memory windows, and the on-state current significantly increases to approximately the same value, while the off-state current gradually increases with increasing temperature, which shows flexible thermal temperature tunability (see Supplementary Figs. S11 and S12a in Supplementary Information for more thermal dependence of the varying VGG range transfer curves). Figure 5a shows the decreasing ON/OFF ratio (red square curve) due to the

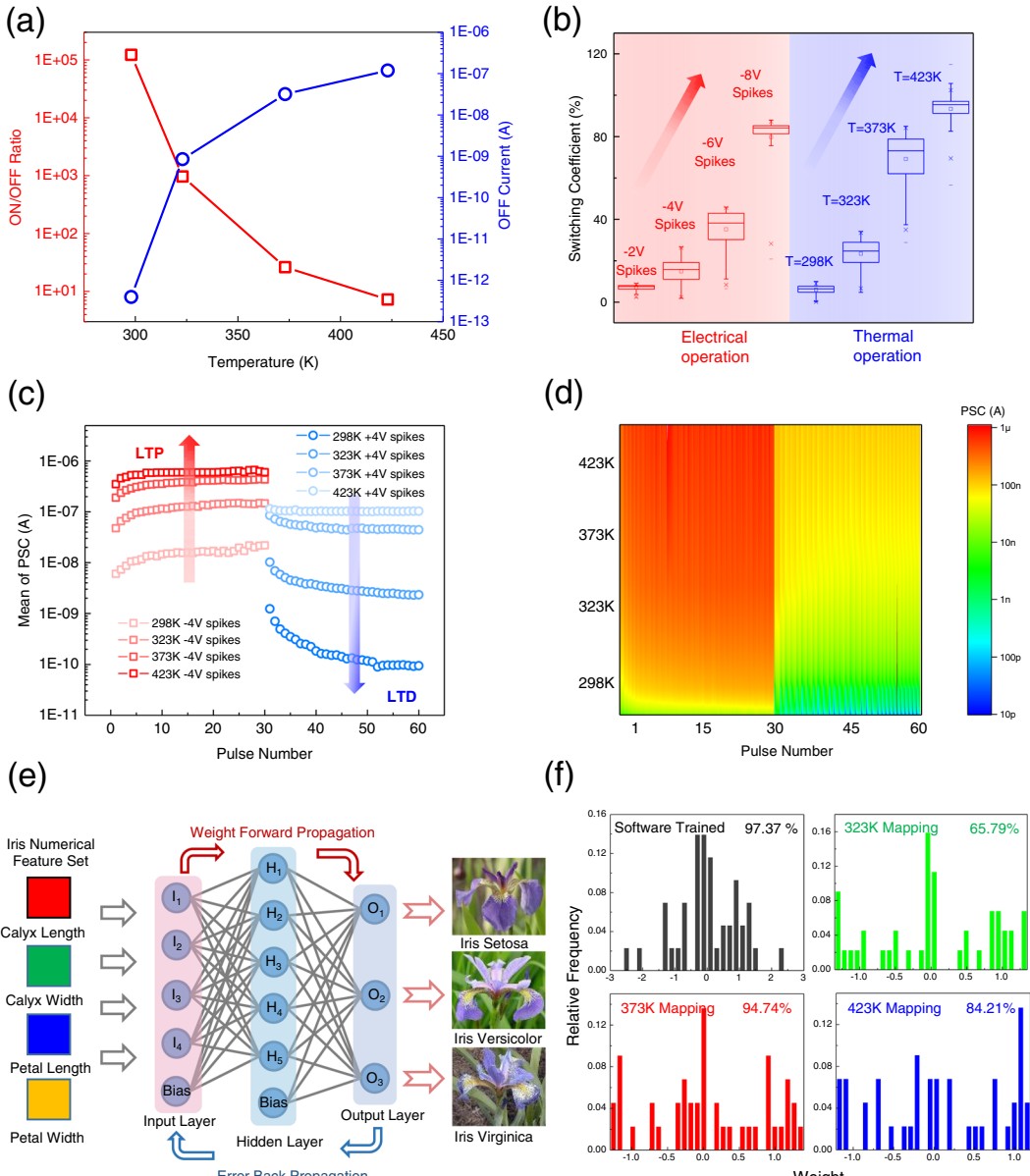

**Fig. 5 Flexible thermal tunable memory switching and neuromorphic computing of 2D α-In₂Se₃ FeCTs. a** ON/OFF ratio and off-state current under thermal temperature modulation, which shows a progressively decreasing ON/OFF ratio (red square curve) and increasing off-state current (blue circular curve). **b** Electrically or thermally modulated switching coefficients, which raise with increasing spike amplitude (red region) or thermal temperature (blue region), indicates the depolarization of α-In₂Se₃ channel ferroelectric and reflects the electrothermal tunability of FeCTs NVM. **c** The mean PSC varies with thermal temperature and spike numbers, which demonstrates the neuroplasticity of LTP and LTD. The increase in PSC is accompanied by an increase in thermal temperature, which corresponds to strengthened LTP, while the PSC in LTD decreases with declining temperature, which implies enhanced inhibition, as shown by the red and blue arrows, respectively. **d** Continuous PSC mapping with thermal temperature and spikes dependence, which successively performs excitatory and inhibitory stimuli. The interface of potentiation and inhibition is clear, and as temperature and spikes accumulate, excitatory PSC increase, while inhibitory PSC do the opposite. **e** The schematic of fully connected neural network based on α-In₂Se₃ FeCTs for iris recognition and classification. **f** The weight distribution and accuracy of simulated fully connected neural network, including software-trained and thermally tuned.

increasing off-state current (blue circular curve) as the thermal temperature increases (see thermal temperature tunable FeCTs NVM LRS and HRS in Supplementary Fig. S12b in Supplementary Information). Furthermore, we define a switching coefficient to demonstrate the electrical and thermal modulation of 2D α-In₂Se₃ FECTS NVM, where the NVM in HRS is refreshed to LRS under the action of electrical or thermal temperatures, and the switching coefficient is obtained compared with the initial LRS. Interestingly, in Fig. 5b (The box chart symbols are consistent

with those described in Fig. 4d), under electrical or thermal operation, the switching coefficient increases with the increase in spike amplitude (red region) or thermal temperature (blue region), respectively, which indicates the depolarization effect of 2D α-In₂Se₃ channel ferroelectrics[18] and reflects the flexible electrothermal tunability of NVM. Besides, we investigated the thermal modulation effect of FeCTs in neural computing. The mean PSC as a function of thermal temperature and the number of spikes (±4 V, 30 ms) is shown in Fig. 5c, which demonstrates

the LTP and LTD neural plasticity in sequence. For LTP, the increase in the mean of PSC is accompanied by a rise in thermal temperature, which corresponds to an enhanced potentiation effect;[10] while for LTD, the mean PSC declines as the thermal temperature drops, which means that the inhibition is strengthened[10], as indicated by the red and blue arrows in Fig. 5c, respectively. A continuous PSC mapping with thermal temperature and spikes dependence is shown in Fig. 5d, in which excitatory and inhibitory spike stimulations are taken successively, the potentiation and depression domains are clearly delimited, and the excitatory PSC increases with the accumulation of thermal temperature and stimulus, and vice versa (Similarly, PSC mapping does not include all temperature ranges).

In addition, thermal tunability is reflected in the simulated fully connected neural network to identify and classify iris flowers on the standard iris flower database as well. For simulation, 5 input neurons correspond to the 4 numerical features of iris, namely, calyx and petal length, width and 1 bias, and 3 output neurons correspond to 3 classes of iris flowers (setosa, versicolor, virginica), as depicted in Fig. 5e. The detailed flow chart of iris recognition and classification based on FeCTs simulated neural network is shown in Supplementary Fig. S13 in Supplementary Information. First, we conduct software training and test to obtain the weight matrix under the optimal recognition accuracy. The hidden layer uses the ReLu activation function, and with summation and activation, hidden layer neurons $H_j$ can be expressed as Eq. 1:

$$H_j = \mathrm{ReLu}\left(\sum_{i=1}^{5} WI_i + b\right)(j = 1, 2, 3, 4, 5, 6) \qquad (1)$$

Where $I_i$ is the input neuron, $W$ is the fully connected weight matrix, and $b$ is the bias. The output value is converted by the Softmax function, followed by fed into the cross-entropy loss function, which can be described as Eq. 2:

$$\mathrm{L}(y_k, \hat{y}_k) = -[y_k \log \hat{y}_k + (1 - y_k) \log(1 - \hat{y}_k)] \qquad (2)$$

Where $y_k$ is the final output of the simulated network, $\hat{y}_k$ is the iris label value. The back propagation algorithm with analog weight update was adopted in the simulation based on α-In$_2$Se$_3$ FeCTs conductance modulation. Subsequently, the thermally tunable FeCTs device conductance is mapped to the optimal weight matrix trained by the software to obtain the mapped network. We assume that any FeCTs conductance level $G$ can be achieved through sophisticated program tuning, then the original software-trained floating-point number can be mapped to the closest device conductance value[37]. The mapping coefficient $λ$ is introduced to scale the available conductance to match the weights trained by the software. The square mapping error (SME) is utilized as the mapping function (Eq. 3), that is, the sum of squares of the difference between the weight pairs before and after conductance mapping[37], which requires finding the optimal $λ$ to minimize SME:

$$\mathrm{SME} = \min\left\{\sum[(1\mathrm{E} + 10)λG - W]^2\right\} \qquad (3)$$

Both thermal temperature and spike amplitudes have an effect on the mapping coefficient, due to the modulation of α-In$_2$Se$_3$ FeCTs channel polarization, which in turn affects SME, as shown in Supplementary Fig. S14a, b in Supplementary Information, respectively. Finally, retest the recognition and classification accuracy of the mapped network after the FeCTs conductance weight matrix mapping. Fig. 5f shows the weight distribution and corresponding accuracy of the software trained and applied thermal modulation conductance mapping. It is worth noting that the highest recognition accuracy of the thermally modulated conductance mapping network reaches 94.74%, which is comparable to the software result (97.37%). And the weight distribution of conductance mapping with varying spike amplitudes at RT is shown in Supplementary Fig. S15 in Supplementary Information. Under thermal modulation, the neural network based on α-In$_2$Se$_3$ FeCTs simulation realizes the iris recognition and classification comparable to software, which presents a potential opportunity for neural network performance optimization.

## Discussion

To summarize, we have demonstrated the 2D α-In$_2$Se$_3$ based FeCTs that integrate ultrafast nonvolatile memory and neural computing functions, which is completely different from conventional FeFETs. As a NVM with large hysteresis windows and long-term robust retention, in addition, thanks to the effect of the internal electric field, the endurance is optimized, and a nonvolatile switching behavior of 40 ns ultra-fast programming is realized. For neural computing, short- and long-term plasticity modulation is implemented, including amplitude-dependent PSC, SRDP, LTP, and LTD. Remarkably, the ultra-low energy consumption of 234/40 fJ per spike for excitation/inhibition is impressive, making it a promising candidate for future energy-efficient memory computing fusion systems. Furthermore, the 2D FeCTs with NVM and neuromorphic computing exhibit flexible thermal tunability, which is essentially that the thermal temperature modulates the polarization of α-In$_2$Se$_3$ channel ferroelectric, and realizing iris recognition and classification simulation with an accuracy of 94.74% comparable to the software. In brief, 2D α-In$_2$Se$_3$ FeCTs as an alternative, provide a promising perspective on building high-density and energy-efficient emerging applications for memory computation integration.

## Methods

**Fabrication of the 2D α-In$_2$Se$_3$ FeCTs**. 30 nm Al$_2$O$_3$ is deposited on the substrate by ALD as the bottom dielectric layer (The I-V and C-V characteristics are shown in Supplementary Fig. S16 in Supplementary Information), the bottom h-BN and 2D α-In$_2$Se$_3$ channel layer is prepared by mechanical exfoliation. Then PVA assists the transfer of h-BN as the top dielectric layer, wet-removes the PVA sacrificial layer, and then uses electron beam lithography to form the electrode pattern. It is worth mentioning that Al$_2$O$_3$ grown by ALD produces electrostatic doping to the α-In$_2$Se$_3$ channel, so the bottom h-BN is needed to optimize the interface. And the transferred top h-BN not only serves as a dielectric, but also provides passivation to the α-In$_2$Se$_3$ channel to isolate the influence of the ambient atmosphere. Finally, source-drain and top gates are deposited by electron beam evaporation, and a heavily doped silicon substrate is used as the global gate.

**Characterization and electrical measurement of the 2D α-In$_2$Se$_3$ FeCTs**. The surface morphology of α-In$_2$Se$_3$ FeCTs was characterized by AFM, showing a typical channel width of 17.5 μm and length of 10 μm. And the thickness of channel α-In$_2$Se$_3$, top and bottom h-BN are about 40, 20 and 20 nm, respectively. To examine the constructed van der Waals heterojunction interface, a cross-sectional analysis of α-In$_2$Se$_3$ FeCTs was performed using a focused ion beam (FIB) system and STEM technology with EDS elements mapping analysis. In addition, the 2D layered materials were characterized by Raman spectroscopy. Channel α-In$_2$Se$_3$ showed strong peaks near 89, 105, 147, 181, 194 cm$^{-1}$, and the peak position at 89 cm$^{-1}$ indicated a hexagonal structure. The Raman spectrum of h-BN showed a peak near 1366 cm$^{-1}$, which corresponds to the in-plane (E$_{2g}$) vibration model. To further confirm the 2H structure, we have supplemented the XRD characterization of α-In$_2$Se$_3$ crystals. The diffraction pattern only shows the c-plane peak and its higher-order interplanar spacing. The channel α-In$_2$Se$_3$ was transferred to a conductive Au/Al$_2$O$_3$ substrate, and the ferroelectric polarization flip was determined by applied PFM tip bias and domain engineering with ±8 V writing bias. The Cascade probe station equipped with Keithley 4200 A semiconductor analyzer was used to characterize the RT electrical properties of α-In$_2$Se$_3$ FeCTs and the simulation of synaptic behavior plasticity under the ambient environment. The thermal tunability of FeCTs NVM and neural plasticity is characterized by the Lakeshore probe station with thermal variable temperature function. And to minimize the interference of ambient light on the properties of the 2D layered material, all electrical measurements are performed in dark conditions.

**Simulation of iris recognition and classification based on FeCTs fully connected neural network**. The realization of iris recognition and classification based on FeCTs simulated neural network includes loading the iris standard data set, defining a fully connected network model through Python PyTorch, training the network, testing the network and obtaining the weight matrix with the optimal accuracy. Subsequently, the device conductance mapping function SME is defined, and the conductance is mapped to the optimal weight matrix using MATLAB to obtain the mapped network. Finally, retest the recognition and classification accuracy of the mapped network after FeCTs conductance mapping.

## Data availability

The data that support the findings of this study are available from the corresponding author upon reasonable request.

## Code availability

The code in MATLAB and Python are available from the corresponding author upon reasonable request.

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

## Acknowledgements

This work was supported by the National Natural Science Foundation of China (61925402, 61851402, 62090032 and 61734003), Science and Technology Commission of Shanghai Municipality (19JC1416600), National Key Research and Development Program (2017YFB0405600), Shanghai Education Development Foundation and Shanghai Municipal Education Commission Shuguang Program (18SG01). The authors thank Fan Wang for her technical support in XRD characterization.

## Author contributions

S.W. designed and conducted the experiments; P.Z. and D.W.Z. conceived the idea; L.L. and Y.D. support the characterization of materials; H.C. and X.H. provided assistance with mechanism analysis and discussion; L.G. and S.M. provide support in neural network simulation and discussion; S.W. wrote the manuscript and all authors contributed to the revision of the manuscript.

## Competing interests

The authors declare no competing interests.
