## [Peer Review File · Nature Communications]

REVIEWER COMMENTS

Reviewer #1 (Remarks to the Author):

This paper presents alpha-In₂Se₃ based ferroelectric channel transistors, demonstrates short/long term potentiation/depression and performs accuracy analysis for iris recognition and classification. Overall the paper presents some novel technical contributions. However, I have the following questions.

1) The authors claim that the presence of mobile charges enhances the retention. I understand the assisting electric field argument. But do the authors have any experimental backing to this claim? Also, can they quantify the retention improvement by comparing their retention characteristics to the ones reported for Ferroelectric transistors ?

2) It is not clear to me if the authors are using spiking neural networks for iris identification. If yes, then how do they use back-propagation for training? Since SNNs deal with spikes, the differentiability of activation function required by back-propagation algorithms needs to be approximated? How do the authors do that? If no (which it seems like since they talk about ReLU activation function), then why do the authors talk in terms of pre-synaptic and post-synaptic spikes in their device characterization. For example, when the authors show long term potentiation, the characterization is performed in terms of number of spikes. There seems to be a disconnect and it is very hard to evaluate the merits of the paper unless this is clarified. The authors should be consistent in terms of their device characterization and the system analysis. If they are talking about spike based potentiation/depression, the system should also be a spiking neural network. On the other hand, if the authors want to evaluate artificial neural networks (ANNs) i.e. non-spiking networks, then the device characterization should be done accordingly.

3) Let me assume that the authors are performing system analysis using ANNs (as per their description of the system evaluation). In that case, the long term potentiation and depression shows high non-linear characteristics, while the authors claim that the accuracy is close to the software based training, which, I believe, assumes ideal linear characteristics. Can the authors quantify the non-linearity and asymmetry in the potentiation/depression characteristics and compare the same to those reported in FeFETs ?

4) The authors are requested to explain the mechanism for short term and long term potentiation/depression and how the same device exhibits these.

5) The energy consumption of 40fJ is the lowest that the device expends. Otherwise for excitation, the lowest energy is 200fJ. I think the author's claim of 40nJ of energy in the abstract is misleading and they should provide the full range of energy.

6) On a similar note, I did not understand why the energy is decreasing for inhibitory synapses as V_g is increase in Fig. 4(c). Please explain more elaborately.

7) I think the analysis of thermal-assisted tunability of memory needs to be supported by some modeling to understand the mechanism. Also, what are the thermal conductivity and other related coefficients of this material ?

8) It will be good to add some information on the Schottky Barrier heights for the source and drain contacts. Is there any contact gating in the proposed device ?

Reviewer #2 (Remarks to the Author):

The authors report the study on 2D ferroelectric α -In₂Se₃ towards the application of non-volatile memory and neural computation. α -In₂Se₃ is recently verified as a new type of ferroelectric semiconductor that is promising to develop nanoelectronics with novel functions. By using α -In₂Se₃ thin films as the channel material in standard field-effect transistor, they demonstrate memory function with good performances such as the long-term retention, ultrafast write speed and ultralow power consumption. With the top dielectric gating, they simulate the short- and long-term biological plasticity. The results are new and will attract broad interests in the scientific community. However, there are several issues required to be well addressed before its further consideration in Nature Communication.

1. The authors use the Raman feature at 89 cm⁻¹ to claim that the α -In₂Se₃ studied follows 2H stacking. However, in a previous work (PRL 120, 227601, 2018) and in reference 30, similar Raman spectrum was observed from 3R structure as evidenced by the HRTEM characterization. The stacking order (2H or 3R) leads to distinct in-plane and out-of-plane electric dipole locking manner between adjacent layers in α -In₂Se₃. This difference affects the in-plane electric polarizations that is important for the planner I-V measurements. Therefore, I suggest that the authors should provide further experimental evidences to support their material structure characterization.
2. For the erase processe in the memory, it is quite confused. As presented in Figure S6a, during the erase process that is to switch the device from HRS to LRS, applying the Erase spike (- 6V) in the device makes the conductance drops and the drain current is three orders of magnitude smaller if compared to the initial HRS state. After the retraction of the erase voltage pulse, the drain current increases to the LRS. Could the authors make further explanations?
3. The authors integrate the memory and the neural computing unit in one single device. When performing the computation, will it induce any polarization decrease that affect the memory function?
4. Some typos need to be corrected in the text, for example, in Page 4 "Then wet transfer h-BN".

Reviewer #3 (Remarks to the Author):

The authors demonstrated the 2D α -In₂Se₃ non-volatile memory and neural computation device to overcome von Neumann bottleneck. Remarkable performance has been demonstrated, including NVM ultra-fast write speed of 40ns, improved endurance through internal electric field, flexible adjustment of neural plasticity, energy consumption as low as 40fJ per event, and thermally modulated 94.74% high-precision iris recognition classification simulation. This prototypical demonstration laid the foundation for an integrated memory computing system with high density and energy efficiency. There are some issues which need to be addressed before considering publishing this manuscript in NC:

1. Some details of the sample used for PFM measurement should be added. What is the thickness of the In₂Se₃? Does the thickness have impact on the switching voltage and speed?
2. In Line 2, Page 2 of Supporting Information, the authors claimed the top hBN layers provide "passivation to the α -In₂Se₃ channel to isolate the influence of the ambient atmosphere." Actually as shown in Figure 1b, there are some parts of In₂Se₃ exposing to air, that is, not being passivated by hBN. The absorption and desorption of the gas molecules may have influence on the operation mechanism of the device. Moreover, the top gate electrode has directly contacted the In₂Se₃ channel, which would lead to a noteworthy large leakage current. So the leakage current should be provided.
3. At the end of Page 4, the authors mentioned that "the TG tunability is not as good as GG". However, the EOT of the top gate dielectric is smaller than that of the global back gate dielectric, as

demonstrated in Ref. 14. The smaller EOT will lead to a more efficient gate turnability. In Line 63-65, Page 5 of Supporting Information, it is not appropriate to directly compare gate voltage between TG and GG. In contrast, the electric field may be more convincing. By the way, the details of the Al₂O₃ layer, i.e., the thickness, the C-V, etc. measurement should be added.

4. Along the same question above, in Figure 3a, the direction of the hysteresis should be marked as in Ref.14, where the hysteresis direction could be determined by EOT has been demonstrated.

5. As mentioned in Page 6, the V_{ds} used for reading is set as 1 V. Is it so large that could polarize the in-plane ferroelectric domains? A test measurement (in-plane PFM or electrical measurement) should be performed to demonstrate the ferroelectric domains would not be polarized at V_{ds} = 1 V, just like that in another In₂Se₃ ferroelectric synaptic transistor report (B. Tang et al., ACS Appl. Mater. Interfaces 2020, 12, 24920-24928).

6. The authors claimed that when the proposed device was written or erased by GG, it was a non-volatile memory. However, according to Figure 3d, Figure S6a and S6b, the current decreases in half within several seconds. Therefore, it is volatile in fact. This phenomenon is illustrated more clearly in Figure S11, in which the HRS increased several orders of magnitudes. How to explain this?

7. In Figure 2e, it is shown that the negative top gate voltage will form a LRS state. Then why the current shown in Figure 4a decreased? And the corresponding description (Line 4, Page7) of "incremental PSC amplitudes" may not be accurate. Actually, it becomes smaller and smaller.

8. In Figure S8b, the authors mentioned in Line 107 that A₂ is lower than A₁. Then how can A₂/A₁ be more than 100%? Actually the amplitude of the second current is smaller than the first one. The vertical coordinate and the formula in Figure S8b are PPF, but the description above the figure is PPD.

9. Every symbol used in the figures should be stated clearly. Especially in Figure 4d and 5b. What do the crosses, dash area, rectangulars and dots stand for?

10. It would be more appropriate to use scattered data points for Figure 4e, 4f and 5d, because the current did not cover all the voltage and temperature value.

11. In Figure 4c, why the inhibitory data are not monotonically smaller as those of excitatory data? Although it is not the main point of this figure, some necessary discussion should not be avoided.

12. In Line 9, Page 8, the authors mentioned "ohmic contact is formed". However, curves in a linear coordinate system would be more convincing. It is recommended to use linear coordinate system for all output characteristic curves.

13. In the caption of Figure S9, the authors attributed the curves becoming symmetrical to the optimization of contact after annealing. It is not enough to draw such a conclusion because it may be due to the thermionic carriers with such a high energy that they can overcome the contact barrier of both sides. If the authors want to demonstrate the results origin from the annealing effect indeed, I-V measurements after heating should be performed.

14. As for iris recognition, why the authors choose the data from the thermal modulated neural computing instead of the electrical modulated neural computing? Is there any differences of the recognition accuracy between thermal modulated data and electrical modulated data?

15. The authors demonstrated both NVM and neural computing applications. Is it possible to link these two applications to a new field?

16. All of the figures may come from the same one proposed device. However, it is vital to show the stability and the repeatability of the device if you want to push it to practical application. In other words, how does the current of the proposed device change after several weeks or months? Also, how many devices have the authors fabricated and how many devices showed the similar characteristics among them?

17. Last but not least, there are some grammar mistakes and typos in overall paper that the authors really should check carefully. For example, spaces are needed between values and units, spaces are also needed between "Figure" and "S5, S7, S8, S9, S13, S14", all of the letters of MATLAB should be capitalized.

Response to Reviewers' Comments

Response to Reviewer 1

General comments:

“This paper presents alpha-In₂Se₃ based ferroelectric channel transistors, demonstrates short/long term potentiation/depression and performs accuracy analysis for iris recognition and classification. Overall the paper presents some novel technical contributions.”

Response:

We thank the reviewer for the approval of our work and valuable suggestions. In general, we have used 2D ferroelectric semiconductor α -In₂Se₃ as the channel material to demonstrate a compact and scalable device that integrates non-volatile memory and neural computing functions. And according to your comments, we have supplemented the experimental evidence of endurance improvement and compared with typical FeFET, explained the specific implementation and system of iris recognition classification simulation, and quantified the non-linearity and asymmetry of the device long-term potentiation and depression characteristics. In addition, we explained the mechanism of short-term and long-term potentiation and depression in the same device, revised the single-spike energy consumption range, theoretically analyzed the thermal modulation of the device, and provided barrier height and contact gating discussions. We have added revisions and highlighted them to improve the readability of the manuscript, and hope that could address all your concerns.

Comment 1:

“The authors claim that the presence of mobile charges enhances the retention. I understand the assisting electric field argument. But do the authors have any experimental backing to this claim? Also, can they quantify the retention improvement

by comparing their retention characteristics to the ones reported for Ferroelectric transistors?”

Response:

Thanks for your valuable comment. We are honored to reach a consensus with the reviewer on the role of the built-in electric field. And **we recognized that the original statement “better retention time” was inaccurate, the existence of a built-in electric field would only allow long retention time**, similar to most current ferroelectric transistors.

The built-in electric field present in FeCTs can improve endurance, as described in the **Abstract** “Remarkable performance includes NVM ultra-fast write speed of 40ns, improved endurance through internal electric field” (page 1 line 18-19). Although the endurance of some ferroelectric transistors have been improved, there are still emerging ferroelectric transistors that have endurance issues (**Ref 1-3**), **especially for HfO₂-based ferroelectric transistors**. After 500 cycles, the LRS/HRS of a typical HfO₂-based ferroelectric transistor was reduced by 25% (**Ref 1**), while our α -In₂Se₃ FeCTs are only degraded by about 5%. **The negligible degradation of LRS/HRS experimental data supports the claim that the built-in electric field improves FeCTs endurance**. In the light of your constructive comments, we have revised the original expression.

The corresponding discussions added in the revised manuscript: “Therefore, positive and negative movable charges are accumulated on the top and bottom surfaces of the channel, forming an upward built-in electric field to maintain the polarization-bound charge, which indicates **long** retention time and improved endurance of NVM.” (page 5 line 14-17)

References:

1. Gong N, Ma T P. A study of endurance issues in HfO₂-based ferroelectric field effect transistors: Charge trapping and trap generation[J]. IEEE Electron Device Letters, 2017, 39(1): 15-18.

2. Zeng B, Liao M, Liao J, et al. Program/Erase Cycling Degradation Mechanism of HfO₂-Based FeFET Memory Devices[J]. IEEE Electron Device Letters, 2019, 40(5): 710-713.
3. Yurchuk E, Müller J, Müller S, et al. Charge-trapping phenomena in HfO₂-based FeFET-type nonvolatile memories[J]. IEEE Transactions on Electron Devices, 2016, 63(9): 3501-3507.

Comment 2:

“It is not clear to me if the authors are using spiking neural networks for iris identification. If yes, then how do they use back-propagation for training? Since SNNs deal with spikes, the differentiability of activation function required by back-propagation algorithms needs to be approximated? How do the authors do that?”

Response:

Thank you for your valuable comment. As mentioned in our manuscript that *“In addition, thermal tunability is reflected in the simulated fully connected neural network to identify and classify iris flowers on the standard iris flower database as well”* (page 10 line 18-19), **We use a fully connected artificial neural networks (not spiking neural networks) for iris recognition and classification simulation.** **Figure R1** shows the detailed flow chart of iris recognition and classification based on FeCTs simulated neural network, including loading the iris standard dataset, defining a fully connected network model, training the network, testing the network, and obtaining the weight matrix under the optimal accuracy. Subsequently, the device conductance mapping function SME is defined (**Ref 1**), the conductance is mapped to the optimal weight matrix, the new network after the conductance mapping is obtained, and the recognition and classification accuracy of the new network is retested.

Figure R1. Flow chart of iris classification based on FeCTs simulated neural network.

(Figure S13)

“If no (which it seems like since they talk about *relu* activation function), then why do the authors talk in terms of pre-synaptic and post-synaptic spikes in their device characterization. For example, when the authors show long term potentiation, the characterization is performed in term of number of spikes. There seems to be a disconnect and it is very hard to evaluate the merits of the paper unless this is clarified. The authors should be consistent in terms of their device characterization and the system analysis. If they are talking about spike based potentiation/depression, the system should also be a spiking neural network. On the other hand, if the authors

want to evaluate artificial neural networks (ANNs) i.e. non-spiking networks, then the device characterization should be done accordingly.”

For 2D α -In₂Se₃ FeCTs, allows the implementation of **ultrafast non-volatile memory** and **bio-realistic plasticity simulations for spiking neural networks (SNNs)**, including spike-amplitude-dependent-plasticity (SADP) (**Figure 4a**), spike-rating-dependent-plasticity (SRDP) (**Figure 4b**), as well as typical voltage-pulse-induced long-term potentiation and depression characteristics (**Figure 4d-f**) for **artificial neural networks (ANNs) applications**. And the spikes may remind the reader of SNNs, while the numerical pulses will be ANNs. In fact, the “spikes” we refer to is the **applied continuous voltage pulse signals**, which is consistent with the description in the widely reported artificial synaptic devices (**Ref 2-3**). **Since the SNN algorithm is not yet mature, we exploit the long-term potentiation and depression characteristics suitable for ANNs to construct iris recognition and classification simulations** as a way to assess the device system application potential.

In the simulation of iris recognition and classification, FeCTs act as an analog synaptic device, and the device conductance is modulated as a network weight. TG voltage pulses simulate presynaptic voltage input and ferroelectric channel current is monitored as post-synaptic current. Based on **the long-term potentiation and depression characteristics of FeCTs obtained by continuous voltage pulses**, we build a fully connected ANNs for iris flower recognition and classification simulation to evaluate the system potential of the proposed α -In₂Se₃ FeCTs. According to your valuable comment, **we have rearranged the Figure 4 device performance characterization to parallelize the plasticity simulations available for SNNs and the long-term potentiation/depression characteristics for ANNs (Figure R2)**, and replaced “spikes” with “pulse number” to avoid misleading.

Figure R2. Neural computation of 2D α -In₂Se₃ FeCTs, including properties available for SNNs and ANNs. (Figure 4)

References:

1. Yan B, Liu C, Liu X, et al. Understanding the trade-offs of device, circuit and application in ReRAM-based neuromorphic computing systems[C]//2017 IEEE International Electron Devices Meeting (IEDM). IEEE, 2017: 11.4. 1-11.4. 4.
2. Kim M K, Lee J S. Ferroelectric analog synaptic transistors[J]. Nano letters, 2019, 19(3): 2044-2050.
3. Si M, Luo Y, Chung W, et al. A Novel Scalable Energy-Efficient Synaptic Device: Crossbar Ferroelectric Semiconductor Junction[C]//2019 IEEE International Electron Devices Meeting (IEDM). IEEE, 2019: 6.6. 1-6.6. 4.

Comments 3:

“Let me assume that the authors are performing system analysis using ANNs (as per their description of the system evaluation). In that case, the long term potentiation and depression shows high non-linear characteristics, while the authors claim that the accuracy is close to the software based training, which, I believe, assumes ideal linear characteristics.”

Response:

We appreciate the constructive comment of the reviewer. And it is recognized that the linearity and symmetry, multiple states and dynamic range of the device

conductance all have a certain impact on the accuracy of network recognition (Ref 1-3). We adopted an ANN for simulation (Figure R3a), and the linearity and symmetry of FeCTs are not perfect, due to the use of identical pulses. Therefore, we have introduced thermal-assisted modulation to help with the accuracy improvement. Thermal assistance can provide a greater dynamic range for network weight mapping, and conductance is selected uniformly throughout the maximum dynamic range (Ref 4), with accuracy at different thermal temperatures of **T=323K: 65.79%**; **T=373K: 94.74%**; **T=423K: 84.21%**. The results showed the best at 373K with improved accuracy and close to software accuracy (Figure R3b).

Figure R3. Iris recognition and classification simulation. (a) A schematic of FeCTs-based fully connected artificial neural network to identify and classify iris flowers. (Figure 5e) (b) Comparison of network recognition accuracy between software training and thermal-assisted FeCTs conductance mapping. (Figure 5f)

References:

1. Liu C, Chen H, Wang S, et al. Two-dimensional materials for next-generation computing technologies[J]. Nature Nanotechnology, 2020, 15(7): 545-557.
2. Yu S. Neuro-inspired computing with emerging nonvolatile memories[J]. Proceedings of the IEEE, 2018, 106(2): 260-285.
3. Zhang W, Gao B, Tang J, et al. Neuro-inspired computing chips[J]. Nature Electronics, 2020, 3(7): 371-382.
4. Yan B, Liu C, Liu X, et al. Understanding the trade-offs of device, circuit and application in ReRAM-based neuromorphic computing systems[C]//2017 IEEE International Electron Devices Meeting (IEDM). IEEE, 2017: 11.4. 1-11.4. 4.

“Can the authors quantify the non-linearity and asymmetry in the potentiation/depression characteristics and compare the same to those reported in FeFETs?”

The non-linearity of the thermal-assisted long-term potentiation and depression characteristics are **3.78/-8.08 (T=323K)**, **2.63/-5.04 (T=373K)**, and **3.26/-6.32 (T=423K)**. And the asymmetry can be calculated by the formula $|\alpha_p - \alpha_d|$, which is 11.86 (T=323K), 7.67 (T=373K), 9.58 (T=423K), respectively. The non-linearity, asymmetry and network accuracy comparisons of representative RRAM (**Ref 4**), FeFET (**Ref 5**), α -In₂Se₃ c-FSJ (**Ref 6**) and thermal-assisted α -In₂Se₃ FeCTs are shown in **Table R1**.

To quantify the non-linearity and asymmetry of the long-term potentiation and depression characteristics of FeCTs, we use the device behavior model in *NeuroSim+* developed by Professor Yu's team to capture the non-linearity α (**Ref 1-3**). First, the long-term potentiation and depression of conductance data are preprocessed to make them mirror-symmetrical and normalize the number of pulses and conductance. Subsequently, the MATLAB script open sourced by Professor Yu's team on GitHub was modified to perform nonlinear fitting, in which **the cycle-to-cycle variation for potentiation and depression were set to 0.015 and 0.025, respectively, and regardless of the device-to-device weight update variation**. Debug to obtain the best fitting curve, and determine the non-linearity for potentiation (α_p) and depression (α_d) by looking up the information table of one-to-one mapping between the non-linearity and the normalized A value open sourced by Professor Yu's team. **FeCTs exhibit tolerable non-linearity and asymmetry, which can be optimized for higher accuracy with thermally assisted modulation.**

Device material	PCMO RRAM	Si Planar FeFET	α -In ₂ Se ₃ c-FSJ	α -In ₂ Se ₃ FeCTs (323K)	α -In ₂ Se ₃ FeCTs (373K)	α -In ₂ Se ₃ FeCTs (423K)
Non-linearity (α_p/α_d)	3.68/-6.76	5.54/-8.08	4.22/-4.22	3.78/-8.08	2.63/-5.04	3.26/-6.32
Asymmetry ($ \alpha_p-\alpha_d $)	10.44	13.62	8.44	11.86	7.67	9.58
Accuracy	~90%	N. A.	~92%	65.79%	94.74%	84.21%
Reference	4	5	6	Our work		

Table R1. The non-linearity, asymmetry and network accuracy comparison of representative RRAM, FeFET, α -In₂Se₃ c-FSJ and thermal-assisted α -In₂Se₃ FeCTs.

References:

1. Chen P Y, Peng X, Yu S. NeuroSim+: An integrated device-to-algorithm framework for benchmarking synaptic devices and array architectures[C]//2017 IEEE International Electron Devices Meeting (IEDM). IEEE, 2017: 6.1. 1-6.1. 4.
2. Yu S, Chen P Y, Cao Y, et al. Scaling-up resistive synaptic arrays for neuro-inspired architecture: Challenges and prospect[C]//2015 IEEE International Electron Devices Meeting (IEDM). IEEE, 2015: 17.3. 1-17.3. 4.
3. Yu S. Neuro-inspired computing with emerging nonvolatile memories[J]. Proceedings of the IEEE, 2018, 106(2): 260-285.
4. Park S, Sheri A, Kim J, et al. Neuromorphic speech systems using advanced ReRAM-based synapse[C]//2013 IEEE International Electron Devices Meeting. IEEE, 2013: 25.6. 1-25.6. 4.
5. Jerry M, Chen P Y, Zhang J, et al. Ferroelectric FET analog synapse for acceleration of deep neural network training[C]//2017 IEEE International Electron Devices Meeting (IEDM). IEEE, 2017: 6.2. 1-6.2. 4.
6. Si M, Luo Y, Chung W, et al. A Novel Scalable Energy-Efficient Synaptic Device: Crossbar Ferroelectric Semiconductor Junction[C]//2019 IEEE International Electron Devices Meeting (IEDM). IEEE, 2019: 6.6. 1-6.6. 4.

Comment 4:

“The authors are requested to explain the mechanism for short term and long term potentiation/depression and how the same device exhibits these.”

Response:

Thank the reviewer for raising the comments. The behavior of single device exhibiting both short- and long-term potentiation and depression has been reported in several papers (**Ref 1-3**), and our team has demonstrated similar phenomena in

previous work (Ref 4). In fact, we have mentioned the working mechanism of short term and long term potentiation/depression in the original manuscript: “GG applies a negative voltage (below coercive voltage), the bottom channel distributes positive polarization bound charges, the energy band bends downward, and accumulates electrons, which forms a low resistance state (LRS “1”), corresponding to the polarization down (Figure 2c).....which is exactly what neural computing expects to simulate the short- and long-term plasticity in biology.” (page 5 line 23-page 6 line 17).

Specifically, **the coverage area of the gate to the channel directly affects the modulation performance**. Although the top dielectric (h-BN) has a lower EOT and induces a stronger electric field (Ref 5), the top electric field does not completely cover the channel, resulting in a weaker polarization of the channel ferroelectric and depolarization in a short time. Channel conductance returns to its initial state, showing short-term plasticity, and negative voltage shows conductance increase (LRS “1”), corresponding to short-term potentiation. However, with the accumulation of negative voltage pulses, the channel ferroelectric polarization is strengthened, and finally a non-volatile polarization is formed, that is, long-term potentiation, and vice versa. **In this way, both short- and long-term potentiation and depression are achieved in our FeCTs**. For a broad readership, we have revised and expanded the original explanation in detail.

The corresponding discussions added in the revised manuscript: “*The working mechanism of TG is similar to GG as exhibited in Figure 2e, f, but it is worth mentioning that **the coverage area of the gate to the channel directly affects the modulation performance.....In this way, 2D FeCTs show the coexistence and evolution of volatile and nonvolatile, which is exactly what neural computing expects to simulate the short- and long-term plasticity in biology.***” (page 6 line 6-17)

References:

1. Huh W, Jang S, Lee J Y, et al. Synaptic Barristor Based on Phase-Engineered 2D Heterostructures[J]. Advanced Materials, 2018, 30(35): 1801447.

2. Zhu J, Yang Y, Jia R, et al. Ion gated synaptic transistors based on 2D van der Waals crystals with tunable diffusive dynamics[J]. *Advanced Materials*, 2018, 30(21): 1800195.
3. Yang C S, Shang D S, Liu N, et al. All-Solid-State Synaptic Transistor with Ultralow Conductance for Neuromorphic Computing[J]. *Advanced Functional Materials*, 2018, 28(42): 1804170.
4. Chen H, Liu C, Wu Z, et al. Time-Tailoring van der Waals Heterostructures for Human Memory System Programming[J]. *Advanced Science*, 2019, 6(20): 1901072.
5. Si M, Saha A K, Gao S, et al. A ferroelectric semiconductor field-effect transistor[J]. *Nature Electronics*, 2019, 2(12): 580-586.

Comment 5:

“The energy consumption of 40fJ is the lowest that the device expends. Otherwise for excitation, the lowest energy is 200fJ. I think the author's claim of 40fJ of energy in the abstract is misleading and they should provide the full range of energy.”

Response:

Thank the reviewer for the kind comments. We agree with the reviewer that **40 fJ is the lowest energy expenditure for inhibitory synapses, while the lowest energy expenditure for excitatory synapses is 200 fJ**. Taking the lowest energy consumption of inhibitory synapses to represent the lowest energy consumption of devices without considering excitatory energy consumption may be misleading. Therefore, we revised the **Figure 4c** (shown in **Figure R4**) and the original description based on the reviewer's suggestions.

The corresponding discussions added in the revised manuscript: *“Remarkable performance includes.....ultra-low energy consumption of 234/40fJ per event for excitation/inhibition,”* (page 1 line 18-page 2 line 1)

“2D α -In₂Se₃ ferroelectric channel transistors (FeCTs) show absorbing performance, including NVM large memory hysteresis windows.....ultra-low power consumption of 234/40fJ per event for excitation/inhibition.” (page 3 line 10-13)

“It is impressive that 2D α -In₂Se₃ FeCTs generally exhibit ultra-low single-spike power consumption, including a minimum of 234fJ for excitatory synapses and a minimum of 40fJ for inhibitory synapses, which makes them promising candidates for energy-efficient neuromorphic systems.” (page 9 line 3-5)

“2D FeCTs show ultra-low power consumption, including the minimum excitation/inhibition of 234/40fJ per spike.” (page 23 line 3-4)

Figure R4. Calculated single event energy consumption of FeCTs for simulated excitatory and inhibitory synapses under varying VG stimuli. (Figure 4c)

Comment 6:

“On a similar note, I did not understand why the energy is decreasing for inhibitory synapses as V_g is increase in Fig. 4(c). Please explain more elaborately.”

Response:

Thanks to the reviewer for the valuable comments. The energy consumption per spike of FeCTs simulated excitatory and inhibitory synapses under different gate voltage stimuli is calculated with the equation: $I \times t_d \times V$, where I , t_d represents the current response and spike duration, V is the source-drain voltage for three-terminal devices (Ref 1). For FeCTs, the current response of excitatory synapses is generally greater than that of inhibitory synapses, and it rises as the voltage amplitude increases, as shown in red in Figure R4. However, for inhibitory synapses, the increase in voltage amplitude leads to a decrease in the response current, and at higher voltage amplitudes, current degradation dominates, which makes the single-spike energy consumption increase first and then decrease, as shown in blue in Figure R4.

Finally, 2D α -In₂Se₃ FeCTs achieved the lowest single-spike power consumption of 234 fJ for excitatory synapses and 40 fJ for inhibitory synapses.

References:

1. Liu C, Chen H, Wang S, et al. Two-dimensional materials for next-generation computing technologies[J]. Nature Nanotechnology, 2020, 15(7): 545-557.

Comment 7:

“I think the analysis of thermal-assisted tunability of memory needs to be supported by some modeling to understand the mechanism. Also, what are the thermal conductivity and other related coefficients of this material?”

Response:

Thank the reviewer for the comments. Based on the semiconductors nature of α -In₂Se₃, **the Fermi-Dirac distribution function**, which represents the probability of electron occupation, enables the interpretation of thermally assisted memory modulation. The saturation channel current (on-state current) is significantly increased to a similar value. When the thermal temperature exceeds a certain value, **it causes a significant increase in the possibility of electrons occupying the conduction band and the off-state current increases sharply (Ref 1)**, which instead leads to a smaller dynamic range. Besides, existing works have shown that the band gap of 2D layered semiconductors decreases with increasing temperature (**Ref 2-3**), which means the increase of high resistance state electrons, the ferroelectric polarization (band bending) caused by the gate electric field is degraded, and the off-state current is significantly increased. Finally, the thermal-assisted tunability of the memory is realized.

As for the thermoelectric performance of the channel material, it has been reported that **the dimensionless thermoelectric figure of merit ZT of α -In₂Se₃ is about 0.23, and the thermal conductivity is 0.28 W m⁻¹K⁻¹ (Ref 4)**.

References:

1. Chen X, Gu L, Liu L, et al. Temperature-switching Logic in MoS₂ Single Transistors[J]. Chinese Physics B, 2020.

2. Plechinger G, Schrettenbrunner F X, Eroms J, et al. Low-temperature photoluminescence of oxide-covered single-layer MoS₂[J]. *physica status solidi (RRL)*–Rapid Research Letters, 2012, 6(3): 126-128.
3. Hannewald K, Stojanović V M, Schellekens J M T, et al. Theory of polaron bandwidth narrowing in organic molecular crystals[J]. *Physical Review B*, 2004, 69(7): 075211.
4. Han G, Chen Z G, Drennan J, et al. Indium selenides: structural characteristics, synthesis and their thermoelectric performances[J]. *Small*, 2014, 10(14): 2747-2765.

Comment 8:

“It will be good to add some information on the Schottky Barrier heights for the source and drain contacts. Is there any contact gating in the proposed device?”

Response:

Thank the reviewer for the constructive comments. **Figure R5a** shows the Arrhenius curve extracted from **Figure S10a** and its linear fitting result, and the slope is obtained. Then draw the extracted slope as a function of V_{ds} , and do a linear fitting to get the intercept S_0 , as shown in **Figure R5b**. **The barrier height Φ_B was calculated to be approximately 450.9 meV.**

The contact gating effect of the metal-2D interface modulated by the back gate, which is relatively common in back-gated 2D devices (**Ref 1-4**), and **it exists in our devices as well**. The current in FeCTs is jointly modulated by back gate and drain voltage, the current under the negative bias drain voltage is greater than the positive bias, and the difference increases incrementally with increasing back gate voltage, **indicating the modulation of the metal-2D interface barrier by back gate, that is, the contact gating effect**. However, by using fixed drain voltage, **the contact gating effect is equivalent to a common base for our results and only affects the current magnitude, which does not affect the functional implementation of FeCTs**. Based on your suggestion, we have added barrier height information in the Supporting Information.

The corresponding discussions added in the Supporting Information: *“As the temperature increases, the current climbs and a better contact is formed (**The calculated barrier height is about 450.9 meV**), which may be attributed to thermally*

affected channel ferroelectric polarization and defect healing.” (page 10 line 8-11 in Supporting Information)

Figure R5. Arrhenius curve and corresponding slope fitting under different V_{ds} . (a) Arrhenius curves of $1000/T$ versus $\ln(I_{ds}/T^{3/2})$, the dotted line is the linear fitting result. (b) The slope of Arrhenius curves extracted under different V_{ds} . And the dashed line is the linear fitting result.

References:

1. Cheng Z, Price K, Franklin A D. Contacting and gating 2-D nanomaterials[J]. IEEE Transactions on Electron Devices, 2018, 65(10): 4073-4083.
2. Prakash A, Ilatikhameneh H, Wu P, et al. Understanding contact gating in Schottky barrier transistors from 2D channels[J]. Scientific reports, 2017, 7(1): 1-9.
3. Chen Z, Appenzeller J. Gate modulation of graphene contacts-on the scaling of graphene FETs[C]//2009 Symposium on VLSI Technology. IEEE, 2009: 128-129.
4. Liu H, Si M, Deng Y, et al. Switching mechanism in single-layer molybdenum disulfide transistors: An insight into current flow across Schottky barriers[J]. ACS nano, 2014, 8(1): 1031-1038.

Response to Reviewer 2

General comments:

“The authors report the study on 2D ferroelectric alpha-In₂Se₃ towards the application of non-volatile memory and neural computation. Alpha-In₂Se₃ is recently verified as a new type of ferroelectric semiconductor that is promising to develop nanoelectronics with novel functions. By using alpha-In₂Se₃ thin films as the channel material in standard field-effect transistor, they demonstrate memory function with good performances such as the long-term retention, ultrafast write speed and ultralow power consumption. With the top dielectric gating, they simulate the short- and long-term biological plasticity. The results are new and will attract broad interests in the scientific community.”

Response:

We thank the reviewer for the approval of our work and valuable suggestions. In general, we have used 2D ferroelectric semiconductor α -In₂Se₃ as the channel material to demonstrate a compact and scalable device that integrates non-volatile memory and neural computing functions. And according to your comments, we have supplemented the experimental evidence of α -In₂Se₃ characterization to determine the crystal structure, described the current response in memory Erase/Write operations, and discussed the impact of neural computation on memory functions. In addition, we have revised and double-checked the typos. We added revisions and highlighting to improve the readability of the manuscript, and hope that could address all your concerns.

Comment 1:

“The authors use the Raman feature at 89 cm⁻¹ to claim that the alpha-In₂Se₃ studied follows 2H stacking. However, in a previous work (PRL 120, 227601, 2018) and in reference 30, similar Raman spectrum was observed from 3R structure as evidenced by the HRTEM characterization. The stacking order (2H or 3R) leads to

distinct in-plane and out-of-plane electric dipole locking manner between adjacent layers in α -In₂Se₃. This difference affects the in-plane electric polarizations that is important for the planner I-V measurements. Therefore, I suggest that the authors should provide further experimental evidences to support their material structure characterization.”

Response:

Thank the reviewer for the valuable comments. **We have supplemented the X-ray Diffraction (XRD) characterization of α -In₂Se₃ crystals used for 2D FeCTs to validate the 2H structure.** We agree with the reviewer that the stacking order (2H or 3R) results in different in-plane and out-of-plane electric dipole locking between adjacent layers in α -In₂Se₃. This difference affects the in-plane electric polarization, which is important for I-V measurement. We have discussed the Raman characterization of the α -In₂Se₃ crystal in the original manuscript: *“Figure 1d shows the Raman spectrum of the channel α -In₂Se₃ to characterize the material properties, which is consistent with previous reports.”* (page 4 line 13-15) It is worth noting that in comparison with the Raman spectra of the 3R structure, **the 2H α -In₂Se₃ crystal always presents an additional splitting peak near 90 cm⁻¹, which can be considered as an indication of 2H stacking (Ref 1).** However, we agree with the reviewer that it is not convincing to claim the 2H structure of α -In₂Se₃ solely based on the peak at 89 cm⁻¹ in the Raman spectrum (as shown in **Figure R1a**). In order to confirm this statement, we have supplemented the XRD characterization of α -In₂Se₃ crystals used for 2D FeCTs. **Figure R1b** shows the diffraction pattern, which **only shows a c-plane peak and its high-ordered interplanar spacing. The peak pattern enables the determination of the lattice constant c (≈ 19.23 Å), which is highly consistent with the reported 2H α -In₂Se₃ and is significantly different from 3R α -In₂Se₃ (Ref 2-3).** In general, the combination of Raman and XRD characterization strongly supports the assignment of our α -In₂Se₃ crystals to the 2H structure. According to your suggestion, we have added the XRD pattern and discussion of α -In₂Se₃ crystal characterization in the revised version.

The corresponding discussions added in the revised manuscript and Supporting Information: “It is worth noting that the additional splitting peak near 90 cm^{-1} can be regarded as an indication of hexagonal (2H) stacking¹⁸. To further confirm the 2H structure, we supplemented the X-ray diffraction (XRD) characterization of $\alpha\text{-In}_2\text{Se}_3$ crystals, as shown in Figure S14.....The peak pattern can determine the lattice constant c ($\approx 19.23\text{ \AA}$), which is consistent with the reported 2H $\alpha\text{-In}_2\text{Se}_3$, and is significantly different from 3R $\alpha\text{-In}_2\text{Se}_3$.” (page 4 line 15-20)

“Figure S2 shows the XRD characterization of $\alpha\text{-In}_2\text{Se}_3$ crystals used for 2D FeCTs, where the diffraction pattern only shows a c -plane peak and its high-ordered interplanar spacing. The peak pattern enables the determination of the lattice constant c ($\approx 19.23\text{ \AA}$), which is highly consistent with the reported 2H $\alpha\text{-In}_2\text{Se}_3$ and is significantly different from 3R $\alpha\text{-In}_2\text{Se}_3$.” (page 3 line 1-4 in Supporting Information)

Figure R1. Raman and XRD characterization of $\alpha\text{-In}_2\text{Se}_3$. (a) Raman characterization of $\alpha\text{-In}_2\text{Se}_3$ (Figure 1d). (b) XRD characterization of the $\alpha\text{-In}_2\text{Se}_3$ crystal, where the diffraction pattern only shows the c -plane peak and its higher-order interplanar spacing (Figure S2).

References:

1. Xue F, Hu W, Lee K C, et al. Room-Temperature Ferroelectricity in Hexagonally Layered $\alpha\text{-In}_2\text{Se}_3$ Nanoflakes down to the Monolayer Limit[J]. Advanced Functional Materials, 2018, 28(50): 1803738.

2. Ho C H, Lin C H, Wang Y P, et al. Surface oxide effect on optical sensing and photoelectric conversion of α -In₂Se₃ hexagonal microplates[J]. ACS applied materials & interfaces, 2013, 5(6): 2269-2277.
3. Jacobs-Gedrim R B, Shanmugam M, Jain N, et al. Extraordinary photoresponse in two-dimensional In₂Se₃ nanosheets[J]. ACS nano, 2014, 8(1): 514-521.

Comment 2:

“For the erase process in the memory, it is quite confused. As presented in Figure S6a, during the erase process that is to switch the device from HRS to LRS, applying the Erase spike (-6V) in the device makes the conductance drops and the drain current is three orders of magnitude smaller if compared to the initial HRS state. After the retraction of the erase voltage pulse, the drain current increases to the LRS. Could the authors make further explanations?”

Response:

Thank you very much for raising the question. As the reviewer mentioned, during the erasing process of switching the device from HRS to LRS, **the instantaneous current** when the Erase spike (-6 V) is applied to the device will drop by about three orders of magnitude. **This is actually the transient response of the device due to the application of a negative voltage pulse, and vice versa.** In fact, for memory testing, we should **pay more attention to the current state after the erase or write voltage pulse is cancelled**, which implies the effect of the erase and program operations on the channel conductance. And as shown in **Figure S7a**, after removing the erase and write pulse operations, the 2D FeCTs exhibit LRS and HRS respectively, and implying significant non-volatile memory characteristics.

Comment 3:

“The authors integrate the memory and the neural computing unit in one single device. When performing the computation, will it induce any polarization decrease that affect the memory function?”

Response:

Thank the reviewer for the valuable comment. We have supplemented the verified experiments with results showing that **performing neural computation will have some impact on the memory performance, but will not cause memory failure, and the memory functions can still be implemented reliably.** As we mentioned that 2D ferroelectric semiconductor α -In₂Se₃ has been served as the channel material to demonstrate a compact and scalable device that **integrates non-volatile memory and neural computing functions.** And we understand what the reviewer is concerned about, namely that since **the memory and computational functions are achieved by adjusting the sole channel ferroelectric polarization** in the same device, which may lead to crosstalk between computation and memory due to polarization degradation. To make sure that the memory capability does not fail as the computation proceeds, we have prepared devices to again test the memory and computational functions.

First we perform a write or erase operation on the device so that the memory is at the initial LRS or HRS, followed by a neural computation operation that applies excitatory and inhibitory pulses to simulate long-term potentiation and depression behavior. And we have explored the effect of the number (**Figure R2a**) and amplitude (**Figure R2b**) of pulses for performing neural computations on the memory state. The results show that **performing neural computations after a predetermined memory state does have an effect on memory performance,** due to computation-induced polarization decay, and increases with increasing number and amplitude of pulses. However, it is worth stating that, **despite the impact of neural computation on memory performance, LRS and HRS still possess sufficient differences (more than one order of magnitude) to allow the memory to operate reliably without failure.** To conclude, performing neural computation will have some impact on the memory performance, but will not cause memory failure, and the memory functions can still be implemented reliably. According to your valuable comment, we have added a discussion of the impact of computation on memory functions in the revised version.

The corresponding discussions added in the revised manuscript: “*And it is worth noting that performing neural computation will have some impact on the memory performance, but will not cause NVM failure, and the memory functions can still be implemented reliably*” (page 9 line 6-8)

Figure R2. The effect of the number and amplitude of neural computational pulses on memory performance. (a) Neural computation has a slight degenerative effect on the memory state (LRS/HRS) and increases with the cumulative number of pulses. (b) Neural computation has a slight degenerative effect on the memory state (LRS/HRS) and increases with larger pulse amplitudes.

Comment 4:

“Some typos need to be corrected in the text, for example, in Page 4 “Then wet transfer h-BN”.”

Response:

Thank you very much for the kind comment. We have revised “wet” to “we”, and double checked for typos.

The corresponding discussions added in the revised manuscript: “*Then we transfer h-BN as the top dielectric layer.....*” (page 4 line 2)

Response to Reviewer 3

General comments:

“The authors demonstrated the 2D α -In₂Se₃ non-volatile memory and neural computation device to overcome von Neumann bottleneck. Remarkable performance has been demonstrated, including NVM ultra-fast write speed of 40ns, improved endurance through internal electric field, flexible adjustment of neural plasticity, energy consumption as low as 40fJ per event, and thermally modulated 94.74% high-precision iris recognition classification simulation. This prototypical demonstration laid the foundation for an integrated memory computing system with high density and energy efficiency.”

Response:

We thank the reviewer for the approval of our work and constructive suggestions. In general, inspired by the potential of 2D ferroelectric in memory and electronic synapse applications reported by Si et al. (Ref 1-2), we have used **2D ferroelectric semiconductor α -In₂Se₃ as the channel material for the first time to demonstrate a compact and scalable device that integrates non-volatile memory and neural computing functions**. According to your comments, we have added the thickness information of α -In₂Se₃ used for PFM and discussed the influence of thickness on switching voltage, coercive voltage field and operating speed. We have discussed the passivation of the top h-BN and leakage current. We have supplemented the thickness of the Al₂O₃ dielectric layer, I-V, C-V characteristics, calculation of EOT and discussion of TG/GG tunability. We have marked the hysteresis direction of the transfer curve, verified that 1 V V_{ds} does not lead to in-plane ferroelectric polarization, and explained the current attenuation phenomenon in the HRS/LRS characteristics. In addition, we have discussed the definition of PPF and corrected the inconsistent description in the original manuscript. We have added the detailed meanings of the symbols in the box chart and explained the reasons for the non-monotonic decrease in the energy consumption of inhibiting synapses. We have replaced all output characteristics with linear coordinate system display, and added

the output characteristics before and after annealing treatment. We have explained the selection of thermally-assisted electrical modulation data for iris recognition and classification, and discussed the potential of linking non-volatile memory and neural computing to future energy-efficient chips and smart systems. Finally, we have added a discussion of device uniformity, repeatability, temporal stability and revised grammatical mistakes and typos. We have added revisions and highlighted them to improve the readability of the manuscript, and hope that the revised version could address all your concerns.

References:

1. Si M, Saha A K, Gao S, et al. A ferroelectric semiconductor field-effect transistor[J]. Nature Electronics, 2019, 2(12): 580-586.
2. Si M, Luo Y, Chung W, et al. A Novel Scalable Energy-Efficient Synaptic Device: Crossbar Ferroelectric Semiconductor Junction[C]//2019 IEEE International Electron Devices Meeting (IEDM). IEEE, 2019: 6.6. 1-6.6. 4.

Comment 1:

“Some details of the sample used for PFM measurement should be added. What is the thickness of the In₂Se₃? Does the thickness have impact on the switching voltage and speed?”

Response:

Thank the reviewer for the valuable comments. The thickness of the α -In₂Se₃ we used for PFM test is about 40nm, which is similar to the channel α -In₂Se₃ used for FeCTs. **Thickness has an effect on the switching voltage (coercive voltage field), the thicker the coercive field decreases, but does not affect the operating speed.**

We have statistically analyzed the switching voltage, coercive voltage (field) and operating speed of α -In₂Se₃ with varying thicknesses (**Ref 1-8**), as shown in **Table R1**. The difference in the coercive voltage field caused by the thickness change is the essence of determining the switching voltage. When the applied voltage (field) is higher than the coercive voltage (field), the polarization flip occurs to realize the

device functions. The results show that as the $\alpha\text{-In}_2\text{Se}_3$ film becomes thicker, the coercive electric field gradually decreases, which is attributed to the weaker depolarization field of the thicker film, and is consistent with the reported work (Ref 4). As for the speed, it depends on the limit of ferroelectric polarization flip, which means that films of different thicknesses enable show a speed of ns. Based on your suggestions, we have supplemented the thickness information of PFM test samples in the revised version.

The corresponding discussions added in the revised manuscript: “Figure 1e shows the three-cycle off-field PFM amplitude hysteresis loop of the 40 nm channel $\alpha\text{-In}_2\text{Se}_3$Figure 1f records the phases of the outer rectangular track and inner square scanned on the same $\alpha\text{-In}_2\text{Se}_3$ by PFM domain engineering with applying +8 V and -8 V bias.....” (page 4 line 23-page 5 line 6)

$\alpha\text{-In}_2\text{Se}_3$ thickness (nm)	Switching voltage (V)	Coercive voltage (V)	Coercive field (V/nm)	ns speed
2.6 (Ref 1)	-3/+3	N. A.	N. A.	N. A.
5 (Ref 2)	-2/+2	N. A.	N. A.	N. A.
6 (Ref 3)	-6/+6 (in-plane)	-6/+6	1	N. A.
8 (Ref 4)	N. A.	-2.64/+2.64	0.33	N. A.
15.3 (Ref 5)	N. A.	-4/+3	0.20	N. A.
20 (Ref 6)	N. A.	+1/+2 (in-plane)	0.1	N. A.
40 (Our work)	-8/+8	-5/+5	0.125	YES
62 (Ref 4)	N. A.	-3.35/+3.35	0.054	N. A.
70 (Ref 7)	-2/+2	0.5~2	0.03	YES
95 (Ref 8)	-4/+4	2~4	0.02	N. A.
120 (Ref 7)	-2.5/+2.5	2~2.5	0.02	YES

Table R1. Statistics of α -In₂Se₃ thickness-dependent switching voltage, coercive voltage field and operating speed.

References:

1. Wan S, Li Y, Li W, et al. Nonvolatile Ferroelectric Memory Effect in Ultrathin α -In₂Se₃[J]. *Advanced Functional Materials*, 2019, 29(20): 1808606.
2. Wan S, Li Y, Li W, et al. Room-temperature ferroelectricity and a switchable diode effect in two-dimensional α -In₂Se₃ thin layers[J]. *Nanoscale*, 2018, 10(31): 14885-14892.
3. Cui C, Hu W J, Yan X, et al. Intercorrelated in-plane and out-of-plane ferroelectricity in ultrathin two-dimensional layered semiconductor In₂Se₃[J]. *Nano letters*, 2018, 18(2): 1253-1258.
4. Io W F, Yuan S, Pang S Y, et al. Temperature-and thickness-dependence of robust out-of-plane ferroelectricity in CVD grown ultrathin van der Waals α -In₂Se₃ layers[J]. *Nano Research*, 2020: 1-6.
5. Si M, Saha A K, Gao S, et al. A ferroelectric semiconductor field-effect transistor[J]. *Nature Electronics*, 2019, 2(12): 580-586.
6. Zhou Y, Wu D, Zhu Y, et al. Out-of-plane piezoelectricity and ferroelectricity in layered α -In₂Se₃ nanoflakes[J]. *Nano letters*, 2017, 17(9): 5508-5513.
7. Si M, Luo Y, Chung W, et al. A Novel Scalable Energy-Efficient Synaptic Device: Crossbar Ferroelectric Semiconductor Junction[C]//2019 IEEE International Electron Devices Meeting (IEDM). IEEE, 2019: 6.6. 1-6.6. 4.
8. Xue F, He X, Retamal J R D, et al. Gate-Tunable and Multidirection-Switchable Memristive Phenomena in a van der Waals Ferroelectric[J]. *Advanced Materials*, 2019, 31(29): 1901300.

Comment 2:

“In Line 2, Page 2 of Supporting Information, the authors claimed the top hBN layers provide “passivation to the α -In₂Se₃ channel to isolate the influence of the ambient atmosphere.” Actually as shown in Figure 1b, there are some parts of In₂Se₃ exposing to air, that is, not being passivated by hBN. The absorption and desorption of the gas molecules may have influence on the operation mechanism of the device.”

Response:

Thank the reviewer for the valuable comments. For our FeCTs, the transfer curve under thermal atmosphere **shows similar device characteristics to that in air**, as shown in **Figure R1a (Figure S12a)**. In other words, **the molecular adsorption of**

the small part exposed on the channel surface has a negligible effect on the operating mechanism of the device, since thermal or vacuum atmosphere enables the desorption process. In short, the top h-BN plays a key role in passivation protection for device performance.

“Moreover, the top gate electrode has directly contacted the In₂Se₃ channel, which would lead to a noteworthy large leakage current. So the leakage current should be provided.”

We have given and discussed the leakage current of the top dielectric (h-BN) in **Figure S8a** of the Supporting Information: *“Figure S8a shows a typical TG transfer curve with V_{ds} of 1 V and V_{GG} of 0 V, where the red curve is the drain current and the black curve is the TG leakage current.....and compared to the channel current, the leakage current is negligible, reflecting the excellent dielectric properties of TG hBN”* (page 8 line 7-11 in Supporting Information). The top gate transfer curves and the corresponding leakage currents of 2D FeCTs are depicted in **Figure R1b (Figure S8a)**, and a small leakage current of about 10 pA can be found, proving that the top gate electrode is not in direct contact with the channel.

Figure R1. GG transfer curve in thermal atmosphere and TG transfer curve with acceptable leakage current at RT. (a) The transfer curves of VGG sweep range of -4 ~ 4 V, -6 ~ 6 V and -8 ~ 8 V at 323 K, and V_{ds} is fixed at 1 V (**Figure S12a**). (b) TG

transfer curve with V_{ds} of 1 V and V_{GG} of 0 V, which shows a significant clockwise hysteresis window and small leakage current (**Figure S8a**).

Comment 3:

“At the end of Page 4, the authors mentioned that “the TG tunability is not as good as GG”. However, the EOT of the top gate dielectric is smaller than that of the global back gate dielectric, as demonstrated in Ref. 14. The smaller EOT will lead to a more efficient gate tunability. In Line 63-65, Page 5 of Supporting Information, it is not appropriate to directly compare gate voltage between TG and GG. In contrast, the electric field may be more convincing.”

Response:

We appreciate the reviewer for the constructive comments. The smaller EOT will result in more efficient gate tunability (**Ref 1**), and the electric field is more convincing. Besides, **the coverage area of the gate to the channel will affect the modulation performance**. Although the EOT of the top gate dielectric (h-BN) is smaller than that of the global gate dielectric in our 2D FeCTs, and it induces a stronger electric field (**Ref 1**), **the top electric field does not completely cover the channel**, resulting in a weaker polarization of the channel ferroelectric and depolarization in a short time. And 2D FeCTs show the coexistence and evolution of volatile and non-volatile under top gate modulation, which corresponds to the short-term and long-term plasticity in biology. While global gate shows individual non-volatility, that is, the control ability of top gate is not as strong as global gate. In short, **the coverage of the channel by the electric field induced by the gate dominates the tunability**. Based on your suggestion, we have revised the original voltage comparison to electric field to make it more convincing.

The corresponding discussions added in the revised Supporting Information: *“However, the drain current of GG modulation is relatively larger than that of TG, which may be due to the lower coverage of the **positive electric field** induced by TG*

(because of the top dielectric layer hBN may break down under a large forward electric field) and the weaker tunability of TG than GG.” (page 6 line 8-11)

References:

1. Si M, Saha A K, Gao S, et al. A ferroelectric semiconductor field-effect transistor[J]. Nature Electronics, 2019, 2(12): 580-586.

“By the way, the details of the Al₂O₃ layer, i.e., the thickness, the C-V, etc. measurement should be added.”

We have performed I-V and C-V tests of the 30 nm Al₂O₃. **Figure R2a** shows the I-V characteristics for negative and positive voltage sweeping, **showing low leakage currents in the Al₂O₃ dielectric layer**. **Figure R2b** shows the C-V characteristics for different voltage sweep ranges, and the inset illustrates the constructed parallel-plate capacitor structure. And we have calculated the dielectric constant of the deposited Al₂O₃ to be about 9. According to your suggestion, we have added a description of the Al₂O₃ dielectric layer to make it more comprehensive.

The corresponding discussions added in the revised manuscript and Supporting Information: *“30 nm Al₂O₃ is deposited on the substrate by atomic layer deposition (ALD) as the bottom dielectric layer, and the bottom h-BN was prepared by mechanical exfoliation for interface optimization, followed by the transfer of the exfoliated 2D α -In₂Se₃ channel layer.” (page 3 line 22-page 4 line 2)*

“30 nm Al₂O₃ is deposited on the substrate by ALD as the bottom dielectric layer (The I-V and C-V characteristics are shown in Figure S16 in Supporting Information), the bottom h-BN and 2D α -In₂Se₃ channel layer is prepared by mechanical exfoliation.” (page 13 line 7-8)

“Figure S16a shows the I-V characteristics for negative and positive voltage sweeping, showing low leakage currents in the Al₂O₃ dielectric layer. Figure S16b shows the C-V characteristics for different voltage sweep ranges, and the inset illustrates the constructed parallel-plate capacitor structure. And we have

calculated the dielectric constant of the deposited Al₂O₃ to be about 9.” (page 16 line 7-10 in Supporting Information)

Figure R2. ALD 30 nm Al₂O₃ I-V and C-V characteristics. (a) The I-V characteristics under positive and negative bias sweeping. (b) The C-V characteristics for different voltage sweep ranges, and the inset illustrates the constructed parallel-plate capacitor structure.

Comment 4:

“Along the same question above, in Figure 3a, the direction of the hysteresis should be marked as in Ref.14, where the hysteresis direction could be determined by EOT has been demonstrated.”

Response:

Thank the reviewer for the kind comments. As for our 2D FeCTs with 30 nm Al₂O₃ gate insulator, the maximum applied voltage is 8 V, and the dielectric constant of Al₂O₃ is calculated in **Comment 3** is about 9, then **EOT \approx 12.99 nm, and the maximum voltage/EOT \approx 0.62 V nm⁻¹**, corresponding to a clockwise hysteresis curve, which is consistent with the reported work (**Ref 1**). According to your suggestions, we have revised the original **Figure 3a** (as shown in **Figure R3**) and supplemented the discussion to make it more comprehensive.

The corresponding discussions added in the revised manuscript: “*Figure 3a shows the 2D α -In₂Se₃ FeCTs transfer curves under varying bidirectional scanning voltages.....and the clockwise hysteresis memory windows enlarge with the incremental GG voltage (VGG) sweeping.....*” (page 6 line 19-21)

Figure R3. Transfer curves of 2D FeCTs NVM, in which the clockwise hysteresis windows expand with increasing VGG, showing cumulative channel polarization (**Figure 3a**).

References:

1. Si M, Saha A K, Gao S, et al. A ferroelectric semiconductor field-effect transistor[J]. Nature Electronics, 2019, 2(12): 580-586.

Comment 5:

“As mentioned in Page 6, the V_{ds} used for reading is set as 1 V. Is it so large that could polarize the in-plane ferroelectric domains? A test measurement (in-plane PFM or electrical measurement) should be performed to demonstrate the ferroelectric domains would not be polarized at $V_{ds} = 1$ V, just like that in another In₂Se₃ ferroelectric synaptic transistor report (B. Tang et al., ACS Appl. Mater. Interfaces 2020, 12, 24920-24928).”

Response:

Thank the reviewer for the valuable comments. We have verified that **the Vds bias voltage of 1 V is not enough to cause the in-plane ferroelectric polarization flip of the channel α -In₂Se₃**. As shown in **Figure R4**, the output characteristics of Vds scanned from -1 V to 1 V under different gate biases **do not show hysteresis**, which is similar to the reported work (**Ref 1**), that is, there is **no resistance switching phenomenon**, and the corresponding **channel in-plane polarization does not occur**. In short, the 1 V Vds bias we use will not cause in-plane polarization of the channel α -In₂Se₃.

Figure R4. FeCTs I-V output characteristics under different gate voltages at RT.

References:

1. Si M, Luo Y, Chung W, et al. A Novel Scalable Energy-Efficient Synaptic Device: Crossbar Ferroelectric Semiconductor Junction[C]//2019 IEEE International Electron Devices Meeting (IEDM). IEEE, 2019: 6.6. 1-6.6. 4.

Comment 6:

“The authors claimed that when the proposed device was written or erased by GG, it was a non-volatile memory. However, according to Figure 3d, Figure S6a and S6b, the current decreases in half within several seconds. Therefore, it is volatile in

fact. This phenomenon is illustrated more clearly in Figure S11, in which the HRS increased several orders of magnitudes. How to explain this?”

Response:

Thank the reviewer for the constructive comments. 2D FeCTs, as a typical non-volatile memory, **have currents determined by the joint contribution of volatile/non-volatile carriers**. The decrease in current within a few seconds after applying an erasure/write voltage pulse is a reflection of the volatile partial carriers, while the clear difference between HRS and LRS currents at steady state is a function of the non-volatile partial carriers.

As we discussed in the original text, **GG applies a negative voltage (erase operation)**, the bottom channel distributes the positive polarization bound charge, which causes the band to bend downward to accumulate electrons, **forming a memory low resistance state LRS**; while **GG applies a positive voltage (write operation)**, and the negatively polarized bound charge is distributed at the bottom of the channel, the energy band is bent upward, and the electrons are exhausted, which leads to a **memory high resistance state HRS**. **The current of FeCTs after applying erase/write voltage pulse is determined by the joint contribution of volatile/non-volatile carrier**. After the voltage field disappears, there is a **dynamic equilibrium process of the carrier relaxation** in the channel for a short period of time (**Ref 1-5**), which shows that the current in the HRS state mentioned by the reviewer increases in a short period of time, corresponding to the volatile part of the carriers. Subsequently, the HRS and LRS currents stabilize and show a significant difference, which corresponds to the effect of non-volatile partial carriers after erase/write voltage pulses are applied. And the dynamic equilibrium process of carriers is commonly found in 2D charge-based non-volatile memory, as shown in **Figure R5a, b**.

However, for judging whether a memory is volatile or non-volatile, the difference between the HRS and LRS current after the erase/write operation should be focused (Ref 6). The long-lived and maintained difference indicates the

effectiveness of the memory erase and write operations, that is, the nonvolatile erase (LRS) and write (HRS) current states, and basically there should be at least one order of magnitude difference between the LRS and HRS currents. In **Figure 3d** and **Figure S7a**, HRS/LRS show significant and persistent differences, indicating that FeCTs are typical of non-volatile memories. And **Figure S12b** demonstrates the depolarization effect of the channel ferroelectric under thermally assisted modulation, enabling a behavior similar to electrical operation.

Figure R5. Reported retention characteristics of 2D nonvolatile memory. (a) The retention result of $WSe_2/MoTe_2$ stacked memory in **Ref 1**. (b) The retention performance of graphene/hBN/MoS₂ devices in **Ref 3**.

References:

1. Park S, Jeong Y, Jin H J, et al. Nonvolatile and Neuromorphic Memory Devices Using Interfacial Traps in Two Dimensional $WSe_2/MoTe_2$ Stack Channel[J]. ACS nano, 2020.
2. Zhang E, Wang W, Zhang C, et al. Tunable charge-trap memory based on few-layer MoS₂[J]. ACS nano, 2015, 9(1): 612-619.
3. Choi M S, Lee G H, Yu Y J, et al. Controlled charge trapping by molybdenum disulphide and graphene in ultrathin heterostructured memory devices[J]. Nature communications, 2013, 4(1): 1-7.
4. Bertolazzi S, Krasnozhan D, Kis A. Nonvolatile memory cells based on MoS₂/graphene heterostructures[J]. ACS nano, 2013, 7(4): 3246-3252.
5. Hou X, Yan X, Liu C, et al. Operation mode switchable charge-trap memory based on few-layer MoS₂[J]. Semiconductor Science and Technology, 2018, 33(3): 034001.

6. Liu C, Yan X, Song X, et al. A semi-floating gate memory based on van der Waals heterostructures for quasi-non-volatile applications[J]. Nature nanotechnology, 2018, 13(5): 404-410.

Comment 7:

“In Figure 2e, it is shown that the negative top gate voltage will form a LRS state. Then why the current shown in Figure 4a decreased? And the corresponding description (Line 4, Page7) of “incremental PSC amplitudes” may not be accurate. Actually, it becomes smaller and smaller.”

Response:

Thank the reviewer for the kind comments. As the reviewer mentioned, the negative top gate voltage will form an LRS state, but the premise is that the global gate bias is 0. However, for the purpose of **obtaining significant neuromorphic spike-amplitude-dependent plasticity (SADP), the global gate bias is fixed at 4V when a short negative top gate spike with increasing amplitude is applied**, which results in an inhibitory PSC, as shown in **Figure 4a**. Besides, we recognized that the incremental PSC amplitude description may be inaccurate. Top gate imposes a short negative spike, as the spike amplitude increases, the FeCTs PSC actually decreases, and **what really increases is the current variations in response to the spike amplitudes**. According to your suggestion, we have revised the original related PSC description to make it accurate.

The corresponding discussions added in the revised manuscript: *“TG applies short negative spikes (-8~-5 V, step 1 V), and with the increase of the spike amplitudes, the FeCTs exhibit **incremental PSC variations** in response to the spikes, but can return to the initial state quickly, which simulates a typical biological short-term plasticity (STP), as shown in Figure 4a.”* (page 7 line 18-21)

Comment 8:

“In Figure S8b, the authors mentioned in Line 107 that A_2 is lower than A_1 . Then how can A_2/A_1 be more than 100%? Actually the amplitude of the second current is smaller than the first one. The vertical coordinate and the formula in Figure S8b are PPF, but the description above the figure is PPD.”

Response:

Thank the reviewer for the valuable comments. **For our FeCTs, it is attributed to the inhibitory current-induced PPF that causes A_2 to be lower than A_1 .** The PPF or PPD index, as a characteristic parameter for evaluating the strength of PPF or PPD, is defined as A_2/A_1 , where **A_1 and A_2 are the absolute amplitudes of excitatory or inhibitory PSC.** Due to the inhibition effect, the PSC after the second pulse is lower than the first. However, **the influence of two consecutive presynaptic pulses on the absolute amplitude of inhibitory PSC is strengthened (Ref 1),** that is, A_2/A_1 exceeds 100%, corresponding to the typical PPF behavior, which is induced by inhibitory PSC. And in recent years, several published works have reported similar inhibitory current-induced PPF characteristics **(Ref 2-3),** as shown in **Figure R6a.** In addition, the PPD characteristics induced by the inhibitory current have been reported **(Figure R6b),** which corresponds to **the weakening of the absolute amplitude of the inhibitory PSC by two consecutive presynaptic pulses (Ref 4).** Therefore, for our FeCTs, it is attributed to the inhibitory current-induced PPF as well. We are very grateful to the reviewer for pointing out the inconsistency in the description caused by our negligence. Based on your kind and valuable comments, we have revised the PPF description in the original manuscript.

The corresponding discussions added in the revised manuscript and Supporting Information: *“Figure S9 investigated the inhibitory PSC induced paired pulse facilitation (PPF) characteristics of STP, which gradually recover to 100% as the spike interval increases, and is described in detail in Supporting Information.”* (page 7 line 22-page 8 line 1)

“Figure S9 describes the characteristics of **inhibitory PSC induced PPF** in neural computing. As the spike interval increases, **inhibitory PSC induced PPF** index gradually returns to 100%, showing the disappearance of the short-term inhibitory effect. The PSC response after paired pulses is shown in Figure S9a. And the PSC amplitude after first spike is recorded as A_1 , and the amplitude after second spike is recorded as A_2 , and when A_2 is lower than A_1 , the synaptic inhibition strengthening effect is simulated. Figure S9b shows the **inhibitory PSC induced PPF** index as a function of interval time, which gradually refresh to 100% as the interval accumulates, corresponding to a typical STP in neural computing.” (page 9 line 9-16 in Supporting Information)

Figure R6. Reported inhibitory current-induced PPF and PPD characteristics. (a) The inhibitory current-induced PPF characteristics in **Ref 2**. (b) The inhibitory current-induced PPD characteristics in **Ref 4**.

References:

1. Sun L, Zhang Y, Hwang G, et al. Synaptic computation enabled by joule heating of single-layered semiconductors for sound localization[J]. Nano Letters, 2018, 18(5): 3229-3234.
2. He H K, Yang R, Zhou W, et al. Photonic potentiation and electric habituation in ultrathin memristive synapses based on monolayer MoS₂[J]. Small, 2018, 14(15): 1800079.
3. Qin S, Wang F, Liu Y, et al. A light-stimulated synaptic device based on graphene hybrid phototransistor[J]. 2D Materials, 2017, 4(3): 035022.

4. Gkoupidenis P, Schaefer N, Garlan B, et al. Neuromorphic functions in PEDOT: PSS organic electrochemical transistors[J]. *Advanced Materials*, 2015, 27(44): 7176-7180.

Comment 9:

“Every symbol used in the figures should be stated clearly. Especially in Figure 4d and 5b. What do the crosses, dash area, rectangulars and dots stand for?”

Response:

Thank the reviewer for the kind comments. We have plotted **box chart** to get **Figure 4d** and **Figure 5b** to show the LTP/LTD characteristics and the switching coefficient under electrical/thermal operation. The box chart is composed of a box body, top and bottom crosses and horizontal lines, and a rectangle, as shown in **Figure R7**. The box body contains data ranging from 25% to 75%. The top and bottom crosses represent the data at 99% and 1% respectively. The top and bottom horizontal lines represent the maximum and minimum values of the data, the rectangle represents the mean value of the data, and the dash area describes the progressive trend of LTP/LTD with the pulses number. According to your suggestion, **we have added a description of the symbols in the box chart** to make it comprehensive.

The corresponding discussions added in the revised manuscript: *“Minimal spikes of ± 0.5 V are applied to the gate.....(The box chart contains data ranging from 25% to 75%, with the cross at the top and bottom representing 99% and 1% of the data, respectively. The upper and lower horizontal lines represent the maximum and minimum values of the data, and the rectangle represents the mean value of the data).”* (page 8 line 10-13)

“Interestingly, in Figure 5b (The box chart symbols are consistent with those described in Figure 4d).....” (page 10 line 2-3)

Figure R7. Schematic and symbol description of box chart used in **Figure 4d** and **Figure 5b**.

Comment 10:

“It would be more appropriate to use scattered data points for Figure 4e, 4f and 5d, because the current did not cover all the voltage and temperature value.”

Response:

We thank the reviewer for the kind comments. In order to visualize the PSC evolution trend under varying gate voltage, thermal temperature and continuous spikes, we have adopted the **2D color filled contour plot** in Origin. Intuitively, we mapped the PSC trend of 10 consecutive spikes with varying gate voltage. Excitatory PSC was obtained under the negative gate spike, and the current increased with the amplitude and spike accumulation, showing **LTP evolution behavior** (as shown in **Figure 4e**). For the inhibitory PSC under the positive spike, the current decreases with the increase in amplitude and spike, showing **typical LTD evolution behavior** (as shown in **Figure 4f**). In addition, we show a continuous PSC mapping with thermal temperature and spike number dependence, in which excitatory and inhibitory spike stimulations are taken successively, the potentiation and depression domains are clearly delimited, and the excitatory PSC increases with the accumulation of thermal temperature and stimulus, and vice versa (as shown in **Figure 5d**). Although the current does not cover all voltage and temperature values, **the color-filled contour mapping enables a more vivid and intuitive perception of the PSC evolution.**

According to your kind comments, **we have added a supplementary note** to the 2D color filled contour plots, where the current in the PSC mapping does not include all voltage and temperature values but to show the evolution trend more vividly and intuitively.

The corresponding discussions added in the revised manuscript: “*Conversely, Figure 4f implies an inhibitory PSC under positive spikes.....(Note that the current in the PSC mapping does not include all voltage ranges, but to show the evolution trend more vividly and intuitively)*” (page 8 line 18-21)

“*A continuous PSC mapping with thermal temperature and spikes dependence is shown in Figure 5d.....and vice versa (Similarly, PSC mapping does not include all temperature ranges)*” (page 10 line 13-17)

Comment 11:

“*In Figure 4c, why the inhibitory data are not monotonically smaller as those of excitatory data? Although it is not the main point of this figure, some necessary discussion should not be avoided.*”

Response:

Thanks to the reviewer for the valuable comments. The energy consumption per spike of FeCTs simulated excitatory and inhibitory synapses under different gate voltage stimuli is calculated with the equation: $I \times t_d \times V$, where I , t_d represents the current response and spike duration, V is the source-drain voltage for three-terminal devices (Ref 1). For FeCTs, the current response of excitatory synapses is generally greater than that of inhibitory synapses, and it rises as the voltage amplitude increases, as shown in red in **Figure R8**. However, for inhibitory synapses, **the increase in voltage amplitude leads to a decrease in the response current, and at higher voltage amplitudes, current degradation dominates**, which makes the single-spike energy consumption increase first and then decrease, as shown in blue in **Figure R9**. Finally, 2D α -In₂Se₃ FeCTs achieved the lowest single-spike power consumption of 234 fJ for excitatory synapses and 40 fJ for inhibitory synapses. According to your

suggestion, we have added discussions on the trend of excitatory and inhibitory synaptic single-spike energy consumption to make it comprehensive.

The corresponding discussions added in the revised manuscript: “*For excitatory synapses, single spike energy expenditure decreases monotonically. As for inhibitory synapses, the increase in voltage amplitude leads to a decrease in the response current, and at higher voltage amplitudes, current degradation dominates, which makes the single-spike energy consumption increase first and then decrease.*” (page 8 line 22-page 9 line 3)

Figure R8. Calculated single-spike energy consumption of FeCTs for simulated excitatory and inhibitory synapses under varying VG stimuli. (Figure 4c)

References:

1. Liu C, Chen H, Wang S, et al. Two-dimensional materials for next-generation computing technologies[J]. Nature Nanotechnology, 2020, 15(7): 545-557.

Comment 12:

“*In Line 9, Page 8, the authors mentioned “ohmic contact is formed”. However, curves in a linear coordinate system would be more convincing. It is recommended to use linear coordinate system for all output characteristic curves.*”

Response:

Thank the reviewer for the kind comments. We have **displayed all output curves** (including GG, TG sweeping and thermally dependent output characteristics) **in a linear coordinate system** to make it more convincing, as shown in **Figure R9**. (**Figure S6, Figure S10a**)

Figure R9. Output curves of different VGG and VTG and thermal-dependent output curves. (a) Output curves under GG modulation, where GG is from -2 to 2 V in 1 V steps. (**Figure S6a**) (b) Output curves under TG modulation, where TG is from -2 to 2 V in 1 V steps. (**Figure S6b**) (c) Thermal temperature-dependent output characteristics, where Vds is from -0.1 to 0.1 V, VGG and VTG are fixed at 0 V. (**Figure S10a**)

Comment 13:

“In the caption of Figure S9, the authors attributed the curves becoming symmetrical to the optimization of contact after annealing. It is not enough to draw such a conclusion because it may be due to the thermionic carriers with such a high energy that they can overcome the contact barrier of both sides. If the authors want to demonstrate the results origin from the annealing effect indeed, I-V measurements after heating should be performed.”

Response:

Thank the reviewer for the valuable comments. We have **annealed the 2D FeCTs at 200 °C in N₂ atmosphere for 2 h**, tested the I-V output characteristics, and compared the performance before annealing. **Figure R10a** shows the output

characteristics of V_{ds} scanned from -1 V to 1 V under different gate biases, which still exhibit **negligible hysteresis after annealing**. **Figure R10b** depicts a comparison of the output characteristics before and after annealing when the gate voltage is 0 V. It shows a significant increase in channel current and improved symmetry, indicating that **the annealing treatment has resulted in an optimized contact**. In conclusion, we demonstrated that the annealing treatment can increase the channel current, enabling the formation of more symmetrical and optimized contacts, which is consistent with the description in the original manuscript.

Figure R10. Output characteristics before and after annealing. (a) 2D FeCTs I-V output characteristics with varying gate voltages after annealing. (b) Comparison of output characteristics before and after annealing.

Comment 14:

“As for iris recognition, why the authors choose the data from the thermal modulated neural computing instead of the electrical modulated neural computing? Is there any differences of the recognition accuracy between thermal modulated data and electrical modulated data?”

Response:

Thank the reviewer for the valuable comments. **We use thermal-assisted electrically modulated conductance data for iris identification and classification**

simulations. The electrical modulation of conductance is the basis, and thermal assistance provides greater conductance dynamic range. And it is recognized that the linearity and symmetry, multiple states and dynamic range of the device conductance will all have a certain impact on the accuracy of the network recognition (**Ref 1-3**).

We adopted an ANN for simulation (**Figure R11a**), and the linearity and symmetry of FeCTs are not perfect, due to the use of identical pulses. Therefore, we **have introduced thermal-assisted modulation to help with the accuracy improvement.** Thermal assistance can provide a greater dynamic range for network weight mapping, and **conductance is selected uniformly** throughout the maximum dynamic range (**Ref 4**), with accuracy at different thermal temperatures of **T=323K: 65.79%; T=373K: 94.74%; T=423K: 84.21%**. The results showed the best at 373K with improved accuracy and close to software accuracy (**Figure R11b**).

Figure R11. Iris recognition and classification simulation. (a) A schematic of FeCTs-based fully connected artificial neural network to identify and classify iris flowers. (**Figure 5e**) (b) Comparison of optimal network recognition accuracy between software training and FeCTs conductance mapping.

References:

1. Liu C, Chen H, Wang S, et al. Two-dimensional materials for next-generation computing technologies[J]. Nature Nanotechnology, 2020, 15(7): 545-557.
2. Yu S. Neuro-inspired computing with emerging nonvolatile memories[J]. Proceedings of the IEEE, 2018, 106(2): 260-285.

3. Zhang W, Gao B, Tang J, et al. Neuro-inspired computing chips[J]. Nature Electronics, 2020, 3(7): 371-382.
4. Yan B, Liu C, Liu X, et al. Understanding the trade-offs of device, circuit and application in ReRAM-based neuromorphic computing systems[C]//2017 IEEE International Electron Devices Meeting (IEDM). IEEE, 2017: 11.4. 1-11.4. 4.

Comment 15:

“The authors demonstrated both NVM and neural computing applications. Is it possible to link these two applications to a new field?”

Response:

Thank the reviewer for the constructive comments. **It is possible to link the non-volatile memory and neural computing to future energy-efficient chips and intelligent system application field.** We have used 2D ferroelectric semiconductor α -In₂Se₃ as the channel material to demonstrate a compact and scalable device that integrates non-volatile memory and neural computing functions. **Future chips tend to develop closer to memory and computing modules, and even use memory devices for computing directly**, which is more compact and energy-efficient than today. And advanced concepts and technologies such as the monolithic 3D integration of memory and logical computing layers (**Ref 1**), near-memory computing (**Ref 2**) and in-memory computing (**Ref 3-5**) have been proposed in recent years. For memory modules, **high-speed** and **low-energy** operation and **non-volatile retention** are required, while for computing modules, **high energy efficiency** and **adaptability** are desired. The prototype 2D FeCTs device we demonstrated realizes **the integration of ultrafast non-volatile memory, high energy efficiency, and tunable neural computing.** And under the collaborative optimization of high-quality large-area ferroelectric channel growth, device arraying and system cascading technology, it is possible to open up an alternative path with 2D FeCTs, which links non-volatile memory and neural computing, to overcome the inefficient separation of memory and computation for future energy-efficient chips and intelligent systems.

References:

1. Wong H S P, Salahuddin S. Memory leads the way to better computing[J]. Nature nanotechnology, 2015, 10(3): 191-194.
2. Lee D U, Kim K W, Kim K W, et al. 25.2 A 1.2 V 8Gb 8-channel 128GB/s high-bandwidth memory (HBM) stacked DRAM with effective microbump I/O test methods using 29nm process and TSV[C]//2014 IEEE International Solid-State Circuits Conference Digest of Technical Papers (ISSCC). IEEE, 2014: 432-433.
3. Ielmini D, Wong H S P. In-memory computing with resistive switching devices[J]. Nature Electronics, 2018, 1(6): 333-343.
4. Chen B, Cai F, Zhou J, et al. Efficient in-memory computing architecture based on crossbar arrays[C]//2015 IEEE International Electron Devices Meeting (IEDM). IEEE, 2015: 17.5. 1-17.5. 4.
5. Sebastian A, Le Gallo M, Khaddam-Aljameh R, et al. Memory devices and applications for in-memory computing[J]. Nature Nanotechnology, 2020: 1-16.

Comment 16:

“All of the figures may come from the same one proposed device. However, it is vital to show the stability and the repeatability of the device if you want to push it to practical application. In other words, how does the current of the proposed device change after several weeks or months? Also, how many devices have the authors fabricated and how many devices showed the similar characteristics among them?”

Response:

Thank the reviewer for the valuable comments. We have fabricated 5 batches of samples with similar conditions, a total of about 25 devices. And **Figures R12a** and **Figure R12b** are the statistical distributions of the maximum values of the memory window (Mean = 5.828, SD = 1.062) and PPF index (Mean = 175.48, SD = 13.38), respectively, which shows that the memory window is 5~6 V in most devices, and the maximum PPF index of most devices can reach 170~180%. Generally, **most devices (76~84%) achieve robust memory and neural computing modulation, and exhibit good repeatability.** Besides, **Figure R13a** and **Figure R13b** show the initial and six-month transfer curves of 2D α -In₂Se₃ FeCTs, respectively. Except for a slight degradation of the saturated on-state current, **the transfer curve trend, memory**

window and threshold voltage are all consistent with the initial state, implying temporal stability of the 2D FeCTs.

Figure R12. Memory window and PPF max index statistics for 2D α -In₂Se₃ FeCTs. (a) Statistical distribution of the maximum memory window. (b) Statistical distribution of the maximum PPF index.

Figure R13. Time-dependent transfer characteristics of 2D α -In₂Se₃ FeCTs. (a) Initial transfer curves. (b) Transfer curves after six months.

Comment 17:

“Last but not least, there are some grammar mistakes and typos in overall paper that the authors really should check carefully. For example, spaces are needed between values and units, spaces are also needed between “Figure” and “S5, S7, S8, S9, S13, S14”, all of the letters of MATLAB should be capitalized.”

Response:

Thank you very much for the kind comment. We have added necessary spaces, corrected the grammar mistakes and double checked for typos.

Based on the reviewers' comments, we have performed the following correction:

Revised Manuscript

1. Page 1, 3 and 12, we have added energy consumption per event for excitation;
2. Page 4 and 14, we have added the discussion of XRD characterization to determine the 2H α -In₂Se₃ structure;
3. Page 6, we have explained the mechanism of achieving both short-term and long-term plasticity under top gate modulation;
4. Page 6, we have added a description of the direction of the hysteresis memory windows;
5. Page 7, we have further explained PPF induced by inhibitory PSC;
6. Page 8, we have added a description of the symbols that appear in the **Figure 4d** box chart, explained the intention of using the PSC mapping, and added a note;
7. Page 9, we have described the evolutionary trend and reasons of single spike energy consumption for excitatory and inhibitory synapses;
8. Page 9, we have supplemented a discussion of the impact of computation on memory functions;
9. Page 10, we have added a note for the box chart symbols in **Figure 5b** and a similar note to the PSC mapping in **Figure 5d**;
10. Page 13, we have added a description of the Al₂O₃ dielectric layer, including the I-V and C-V characteristics;
11. Page 15, we have supplemented a new additional acknowledgment;
12. Page 22, we have rearranged the **Figure 4** to parallelize the plasticity simulations available for SNNs and the characteristics for ANNs;
13. Page 27, we have added 3 new references about 2D α -In₂Se₃ crystal structure.

Revised Supporting Information

1. Page 3, we have added XRD characterization of α -In₂Se₃ as **Figure S2**;
2. Page 6, we have discussed the modulation of top gate and global gate from the perspective of electric field;
3. Page 9, we have revised the description of PPF, which is induced by inhibitory PSC;
4. Page 10, we have supplemented the barrier height information of the 2D FeCTs;
5. Page 16-17, we have added the I-V and C-V characteristics of ALD 30 nm Al₂O₃;
6. Page 17, we have added 2 new references about 2D α -In₂Se₃ crystal structure.

REVIEWERS' COMMENTS

Reviewer #1 (Remarks to the Author):

Thanks to the authors for addressing my comments adequately.

Reviewer #2 (Remarks to the Author):

The authors have addressed all the questions, and I recommend its publication on Nature Communications.

Reviewer #3 (Remarks to the Author):

The authors have answered reviewers' questions properly and revised the manuscript accordingly. I would like to recommend to accept it for publication.

Response to Reviewers' Comments

Response to Reviewer 1

General comments:

"Thanks to the authors for addressing my comments adequately."

Response:

We thank the reviewer for the approval of our work and valuable suggestions.

Response to Reviewer 2

General comments:

"The authors have addressed all the questions, and I recommend its publication on Nature Communications."

Response:

We thank the reviewer for the approval of our work and valuable suggestions.

Response to Reviewer 3

General comments:

"The authors have answered reviewers' questions properly and revised the manuscript accordingly. I would like to recommend to accept it for publication."

Response:

We thank the reviewer for the approval of our work and valuable suggestions.